# Enhanced oxygen reduction with single-atomic-site iron catalysts for a zinc-air battery and hydrogen-air fuel cell

Yuanjun Chen[1], Shufang Ji[1], Shu Zhao[2], Wenxing Chen[1], Juncai Dong [3], Weng-Chon Cheong[1], Rongan Shen[1], Xiaodong Wen[4], Lirong Zheng[3], Alexandre I. Rykov[5], Shichang Cai[6], Haolin Tang[6], Zhongbin Zhuang[7], Chen Chen[1], Qing Peng[1], Dingsheng Wang [1] & Yadong Li[1]

Efficient, durable and inexpensive electrocatalysts that accelerate sluggish oxygen reduction reaction kinetics and achieve high-performance are highly desirable. Here we develop a strategy to fabricate a catalyst comprised of single iron atomic sites supported on a nitrogen, phosphorus and sulfur co-doped hollow carbon polyhedron from a metal-organic framework@polymer composite. The polymer-based coating facilitates the construction of a hollow structure via the Kirkendall effect and electronic modulation of an active metal center by long-range interaction with sulfur and phosphorus. Benefiting from structure functionalities and electronic control of a single-atom iron active center, the catalyst shows a remarkable performance with enhanced kinetics and activity for oxygen reduction in both alkaline and acid media. Moreover, the catalyst shows promise for substitution of expensive platinum to drive the cathodic oxygen reduction reaction in zinc-air batteries and hydrogen-air fuel cells.

[1] Department of Chemistry, Tsinghua University, 100084 Beijing, China. [2] Beijing Guyue New Materials Research Institute, Beijing University of Technology, 100124 Beijing, China. [3] Beijing Synchrotron Radiation Facility, Institute of High Energy Physics, Chinese Academy of Sciences, 100049 Beijing, China. [4] State Key Laboratory of Coal Conversion, Institute of Coal Chemistry, Chinese Academy of Sciences, 030001 Taiyuan, China. [5] Mössbauer Effect Data Center, Dalian Institute of Chemical Physics, Chinese Academy of Sciences, 116023 Dalian, China. [6] State Key Laboratory of Advanced Technology for Materials Synthesis and Processing, Wuhan University of Technology, 430070 Wuhan, China. [7] State Key Lab of Organic-Inorganic Composites and Beijing Advanced Innovation Center for Soft Matter Science and Engineering, Beijing University of Chemical Technology, 100029 Beijing, China. These authors contributed equally: Yuanjun Chen, Shufang Ji, Shu Zhao. Correspondence and requests for materials should be addressed to D.W. (email: wangdingsheng@mail.tsinghua.edu.cn)

Efficient fuel cells and metal-air batteries are limited by sluggish kinetics and high overpotentials of the cathodic electrochemical oxygen reduction reaction (ORR), which has been a bottleneck for the implementation of these energy technologies[1-3]. Although Pt-based materials have served as the most efficient catalysts for ORR, the high cost and scarcity of precious metal platinum as well as the issue of methanol cross-over have motivated the exploration of efficient and durable non-noble metal catalysts[4-9]. Among them, metal-nitrogen-carbon (M-N-C) catalysts have been regarded as promising alternatives to Pt-based materials[10-12]. However, their catalytic performance is still far from satisfactory, partly because active sites are not primarily and preferentially formed during synthesis[13,14]. Most synthetic routes to M-N-C catalysts necessitate high-temperature pyrolysis, which often leads to the coexistence of active species and a large amount of less-active metal particles or carbide phases[15]. Such heterogeneity in structure and composition not only contributes to unsatisfactory performance due to the low number of active sites, but also hinders an in-depth understanding of active sites and further establishment of definitive correlation with catalytic properties[13,16,17].

The performance of catalysts depends on rational design and optimization of their structural and electronic properties. Downsizing active species of M−N−C catalysts to single-atom scale can promote maximum atom-utilization efficiency and make active sites fully exposed, which can enhance intrinsic nature of catalysts[18-23]. It is well accepted that doping with heteroatoms within the skeleton of a carbon matrix can efficiently improve electronic features and electrical conductivity of catalysts[24-29]. Modifying the electronic structures of active centers is a powerful approach to enhance catalytic properties[30-33]; however, it is difficult to achieve merely through introducing heteroatoms due to poor control over dispersion and uniformity of dopant heteroatoms. Constructing hollow structures with hierarchical pore distribution to enhance substrate structure functionalities is another effective approach to boost catalytic performance because it benefits the accessibility of active sites and the mass transport properties[34-36]. The ORR catalytic efficiency and kinetics are correlated with multiple steps, including the adsorption and activation of substrates, charge transfer, and desorption of products[4,37]. Remarkable ORR enhancement is difficult to achieve by optimizing only one aspect of a catalyst. Therefore, developing an effective synthetic strategy to preferentially generate uniform and atomically dispersed active sites, and simultaneously achieve electronic modification and structure functionalities is highly desirable but remains challenging.

Here, a novel strategy is developed to construct a functionalized hollow structure from a metal-organic framework (MOF)@polymer via Kirkendall effect and achieve electronic modulation of an active center by near-range coordination with nitrogen and long-range interaction with sulfur and phosphorus. The designed catalyst comprised of single iron atomic sites supported on a nitrogen, phosphorus and sulfur co-doped hollow carbon polyhedron (Fe-SAs/NPS-HC) exhibits superior ORR performance in alkaline media with a positive half-wave potential ($E_{1/2}$) of 0.912 V vs. reversible hydrogen electrode (RHE), the highest kinetic current density ($J_k$) of 71.9 mV cm$^{-2}$ at 0.85 V, and a Tafel slope of 36 mV dec$^{-1}$ that is record-level among previously reported ORR catalysts, to the best of our knowledge. Outstanding catalytic performance for ORR in acidic media is also achieved in Fe-SAs/NPS-HC with an $E_{1/2}$ of 0.791 V, which approaches that of the Pt/C and surpasses that of most reported nonprecious metal catalysts. Furthermore, it possesses outstanding methanol tolerance and electrochemical stability. The control experiments reveal that the designed hollow structure plays an important role in accelerating ORR kinetics and enhancing performance. Density

functional theory (DFT) calculations demonstrate that high efficiency and satisfactory kinetics of Fe-SAs/NPS-HC are attributed to atomic dispersion of N-coordinated Fe and electronic effects from the surrounding S and P atoms, which can donate electrons to single-atom Fe centers to make the charge of Fe (Fe$^{\delta+}$) less positive to weaken the binding of adsorbed OH species. Moreover, Zn-air battery and H$_2$-air fuel cell tests demonstrate that Fe-SAs/NPS-HC exhibits competitive performance compared with Pt/C, suggesting its potential application in energy storage and conversion devices.

## Results

**Synthesis and characterization.** The synthetic route for Fe-SAs/NPS-HC is illustrated in Fig. 1a. The monomers of poly(cyclotriphospazene-co-4,4′-sulfonyldiphenol) (PZS) and iron precursors were mixed with a zeolitic imidazolate framework (ZIF-8) to form ZIF-8/Fe@PZS core-shell composites via polymerization. A transmission electron microscopy (TEM) image (Fig. 1b) shows that ZIF-8/Fe@PZS sample displays a polyhedral morphology with uniform size. A high-resolution TEM (HRTEM) image of ZIF-8/Fe@PZS reveals that PZS is uniformly coated on the surface of ZIF-8 with shell thickness of 20 nm (inset of Fig. 1b). As observed in energy-dispersive spectroscopy (EDS) mappings, Fe, C, N, P, and S elements are uniformly distributed throughout the entire coating layer (Supplementary Figure 1). The final catalyst Fe-SAs/NPS-HC was obtained by pyrolyzing ZIF-8/Fe@PZS at 900 °C in Ar.

TEM and high angle annular dark field scanning TEM (HADDF-STEM) images show that Fe-SAs/NPS-HC has a uniform hollow morphology (Fig. 1c, d). The hierarchical porous structure of Fe-SAs/NPS-HC is revealed by the pore-size distribution plots (Supplementary Figure 2). A powder X-ray diffraction (PXRD) pattern of Fe-SAs/NPS-HC shows one broad peak at about 25° ($2\theta$) assigned to graphitic carbon and no obvious signals for metallic Fe species are detected (Supplementary Figure 3). The content of Fe is determined as 1.54 wt% by inductively coupled plasma optical emission spectrometry (ICP-OES). The homogeneous spatial distribution of Fe, C, N, P, and S elements throughout the entire hollow shell is evidenced by EDS mappings (Fig. 1e). Furthermore, atomic dispersion of Fe is directly observed and confirmed by aberration-corrected HAADF-STEM (AC HAADF-STEM) analysis (Fig. 1f, g), which shows some individual bright dots (marked with yellow circles), corresponding to isolated single Fe atoms.

The binding states of C, N, P, and S in Fe-SAs/NPS-HC were investigated by X-ray photoelectron spectroscopy (XPS) in Supplementary Figure 4. The C 1s spectrum can be deconvoluted into four peaks at the binding energy of 288.3, 285.6, 284.5, and 284.8 eV, corresponding to C−N, C−P, C−S, and C=C, respectively (Supplementary Figure 4a). The N 1s spectrum reveals the coexistence of four types of nitrogen species, pyridinic N (398.7 eV), pyrrolic N (400.3 eV), graphitic N (401.3 eV) and pyridinic N$^+$-O$^-$ (403.7 eV) (Supplementary Figure 4b). The P 2p spectrum displays two peaks located at 132.8 and 133.9 eV, indexing to P−C and P−O (Supplementary Figure 4c). Supplementary Figure 4d shows that the S 2p spectrum can fit well with three peaks at 164.0, 165.2, and 168.3 eV, assigned to 2p3/2, 2p1/2 splitting of the S 2p spin orbital (−C−S−C−) and oxidized S, respectively.

**Formation process of functionalized hollow structure.** To understand the formation mechanism of the hollow structure of Fe-SAs/NPS-HC, a series of control experiments were carried out. Firstly, we investigated the effect of the PZS shell layer. As verified by thermogravimetric analysis in Supplementary Figure 5, the onset

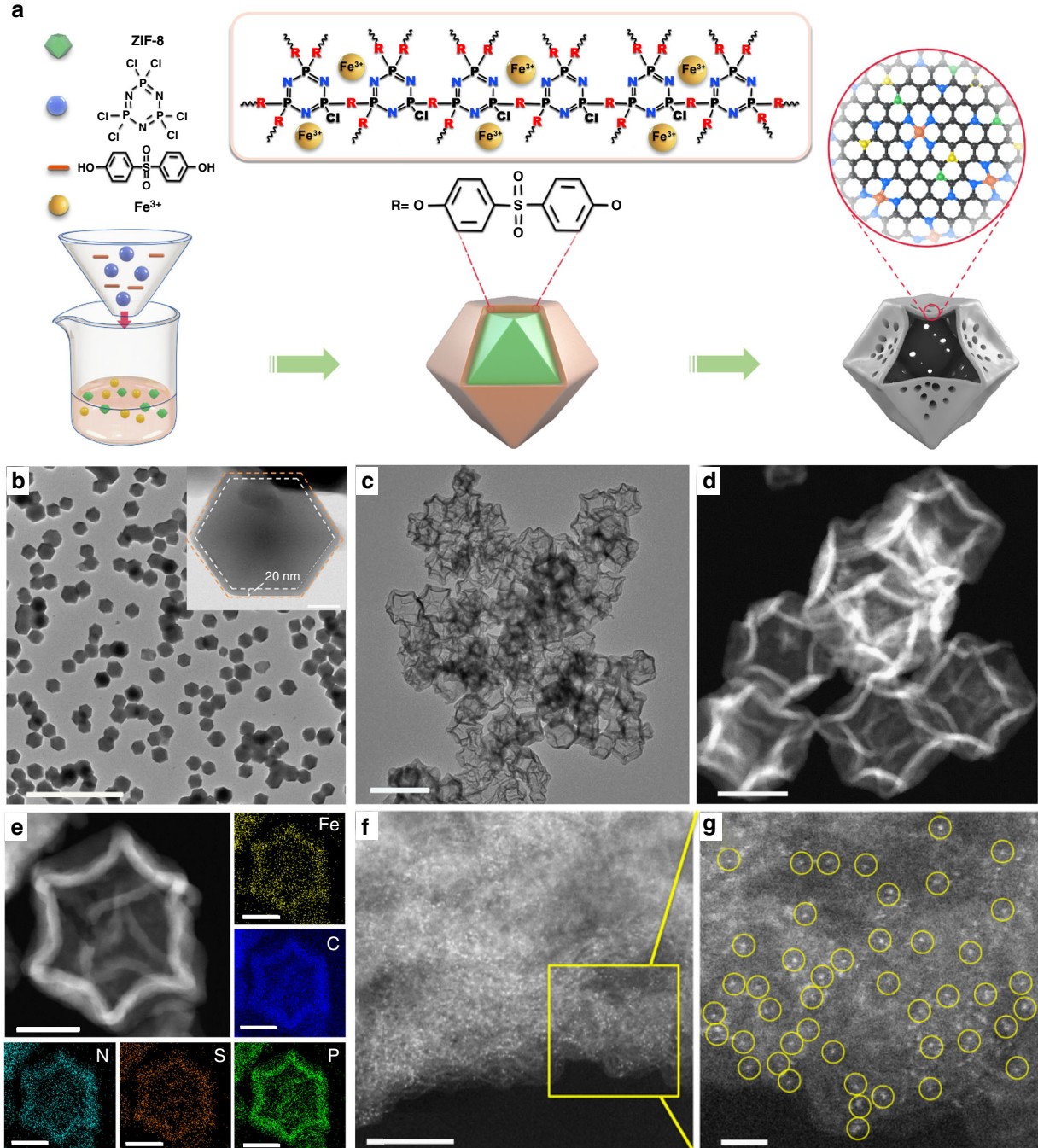

**Fig. 1** Synthesis and structural characterizations. **a** Illustration of preparation process of single iron atomic sites supported on a nitrogen, phosphorus and sulfur co-doped hollow carbon polyhedron (Fe-SAs/NPS-HC). **b** Transmission electron microscopy (TEM) image of Fe-SAs/NPS-HC predecessor, marked as zeolitic imidazolate framework-8 (ZIF-8)/Fe@poly(cyclotriphospazene-co-4,4′-sulfonyldiphenol) (PZS). Scale bar, 2 μm. Inset: high-resolution TEM (HRTEM) image of ZIF-8/Fe@PZS, indicating that the thickness of the PZS shell is approximately 20 nm; scale bar, 100 nm. **c** TEM image of Fe-SAs/NPS-HC. Scale bar, 500 nm. **d** High angle annular dark field scanning TEM (HAADF-STEM) image of Fe-SAs/NPS-HC. Scale bar, 200 nm. **e** The enlarged HAADF-STEM image and corresponding element maps (Fe: yellow, C: blue, N: cyan, S: orange, P: green). Scale bar, 100 nm. **f, g** Aberration-corrected (AC) HAADF-STEM image and enlarged image of Fe-SAs/NPS-HC catalyst. Scale bar, 5 nm (**f**); 1 nm (**g**)

of decomposition of ZIF-8/Fe@PZS occurs at about 400 °C, which is lower compared with that of pure ZIF-8 at 550 °C. This result reveals the PZS layer can induce decomposition of ZIF-8, as further confirmed by the obviously different degrees of attenuation of characteristic XRD peaks of ZIF-8 (Supplementary Figure 6). Furthermore, the designed sample ZIF-8/Fe@PZM was synthesized by the same preparation process as for ZIF-8/Fe@PZS except for changing 4,4′-sulfonyldiphenol into bis(4-aminophenyl) ether

(see details), where no S element is contained. Single iron atomic sites supported on nitrogen and phosphorus co-doped carbon polyhedron (Fe-SAs/NP-C) and nitrogen-doped carbon (N-C) catalysts were obtained by pyrolysis of ZIF-8/Fe@PZM and ZIF-8. As shown in Supplementary Figure 7, the morphology of Fe-SAs/NP-C and N-C is similar with a porous solid structure, indicating the critical role of sulfur for the formation of the hollow structure. Finally, we collected and characterized the intermediates at different

pyrolysis stages at 400, 500 and 600 °C to track the evolutionary trajectory of hollow structure, as schematically illustrated in Fig. 2a based on TEM observations (Fig. 2b–d). At the pyrolysis temperature of 400 °C, the plane surfaces of ZIF-8 of ZIF-8/Fe@PZS are partially etched (Fig. 2b), leading to the emergence of concave morphology with rough and shrinking surfaces, while their edges still maintain pristine structure. By pyrolysis at 500 °C, the surfaces and interior region of ZIF-8 are further removed away and the framework structure with hollow interior and interconnecting edges is formed, as evidenced by TEM image in Fig. 2c. The construction of framework provides a support to prevent the collapse and fracture of shell layers during the evolution process of hollow structure. Then, the edges of ZIF-8 are gradually etched out and thick hollow structure is obtained at 600 °C (Fig. 2d). To more clearly display the morphology and spatial distribution of elements, the corresponding HAADF-STEM images and EDS mappings in various stages are shown in Fig. 2e–g, which strongly supports the above analysis. This process may follow the Kirkendall effect[38]. $S^{2-}$ ion from PZS shell induces decomposition of ZIF-8 for releasing $Zn^{2+}$ ion. Because of smaller ionic radius of $Zn^{2+}$ ion compared with $S^{2-}$ ion, the outward spread of $Zn^{2+}$ ion is faster than the inward transport of $S^{2-}$ ion. Then, the continuous unequal interdiffusion of the $S^{2-}$ ion and $Zn^{2+}$ ion in the interface of ZIF-8 core and PZS shell results in the emergence of Kirkendall voids. As Kirkendall effect process goes on, the inner ZIF-8 gradually decomposes and forms a thick hollow shell structure. When the pyrolysis temperature continues to be elevated and maintains at 900 °C for 3 h, residual Zn in thick hollow shell vaporizes and escapes, resulting in the construction of final hollow structure of Fe-SAs/NPS-HC.

**Atomic structure analysis**. To investigate the local structure of Fe-SAs/NPS-HC, X-ray absorption fine structure (XAFS) measurements were performed. As shown in Fig. 3a, the absorption edge of X-ray absorption near-edge structure (XANES) spectroscopy of Fe-SAs/NPS-HC is situated between those of Fe foil and $Fe_2O_3$ reference, elucidating that isolated Fe atoms carry partially positive charges (Supplementary Figure 8). As shown in Fig. 3b, the Fourier transform (FT) curve of extended X-ray absorption fine structure (EXAFS) of Fe-SAs/NPS-HC only presents Fe−N coordination with a peak at about ~1.5 Å (without phase correction) and an Fe−Fe path at ~2.2 Å is not detected, which clearly indicates that Fe atoms are atomically dispersed and stabilized by nitrogen. Furthermore, wavelet transform (WT) analysis was carried out, which can discriminate the backscattering atoms and provide powerful resolution in $k$ space and $R$ space[39]. As shown in Fig. 3c, the WT contour plot of Fe-SAs/NPS-HC exhibits only one intensity maximum at approximately 4.2 Å$^{-1}$, ascribed to Fe−N contribution. And no intensity maximum indexed to Fe−Fe coordination can be observed compared with the WT contour plot of Fe foil in Fig. 3c. The above results well confirm that Fe species exist as isolated atoms in Fe-SAs/NPS-HC.

In order to further confirm the Fe state of Fe-SAs/NPS-HC, we performed Mössbauer spectroscopy measurements, which reveal the spectra with isomer shift (IS) of $Fe^{3+}$ or low/intermediate spin state of $Fe^{2+}$. Mössbauer spectra of Fe-SAs/NPS-HC at 292 K and at 78 K are presented in Supplementary Figure 9, and the corresponding hyperfine parameters are listed in Supplementary Table 1. The room temperature (RT) spectra are clearly of relaxational nature revealed by slowly decaying wings (Supplementary Figure 9a). From the RT-spectra fitted with a single asymmetric component, implying either magnetic Blume[40] or electron-hopping relaxation mechanisms[41], the average IS of 0.48 and 0.45 mm s$^{-1}$ could be determined, which are clearly higher than the IS of high-spin $Fe^{3+}$. Measurements at 78 K revealed

indeed the absence of high-spin $Fe^{3+}$ (Supplementary Figure 9b). Two magnetic sextet subspectra evolved at 78 K from the relaxational RT-spectra were assigned to $Fe^{III}N_4$ ($S = 1/2$) and $Fe^{II}N_4$ ($S = 1$) species identified by the hyperfine fields ($B_{hf}$) of 10.5(1) T and 17.3(1) T, respectively, according to the general rule for the saturation fields $B^{sat}_{hf}/S \approx 20$ T. Ferromagnetic ordering in the dilute system of $S = 1/2$ and $S = 1$ species below Curie temperature ($T_C$) of 215 K was observed in the temperature dependence of magnetic susceptibility (Supplementary Figure 10) measured at the applied field of 1 kOe and in M−H loops at 2, 77 and 300 K (Supplementary Figure 11). Supplementary Figure 12 shows indeed that the measured $B_{hf}$ are close to saturation $B_{sat}$ (at 0 K). A minor (20%) doublet species was identified with low-spin $Fe^{2+}$, because the signal from single-atom dispersed diamagnetic $S = 0$ species may only experience the line broadening, but not Zeeman splitting below $T_C$. The average oxidation state of Fe in Fe-SAs/NPS-HC is near +2.5. The axial orientations of both fields $B_{hf}$ (parallel to Vzz) reveal that only $Fe^{III}N_4$ and $Fe^{II}N_4$ species are confirmed, and no other Fe-related phases (such as Fe, $Fe_xC$, $Fe_xS$, and $Fe_xP$) are detected, indicating that isolated Fe atoms are only coordinated by N atoms, consistent with the XAFS analysis.

According to the EXAFS fitting curve in Fig. 3d and fitting parameters in Supplementary Table 2, the best-fitting results clearly demonstrate that the first shell peak at 1.5 Å is ascribed to isolated Fe atoms coordinated four N atoms as Fe−N$_4$ structure in comparison with the fitting results for Fe foil and $Fe_2O_3$ (Supplementary Figures 13 and 14). Based on the hyperfine parameters of Mössbauer spectra and the best-fitting EXAFS results as well as the element content analysis by XPS (Supplementary Figure 15), DFT calculations were used to construct and optimize the structural model of Fe-SAs/NPS-HC (Fig. 3e).

**Electrocatalytic performance**. To evaluate ORR performance of the as-prepared catalysts, rotating disk electrode (RDE) measurements in $O_2$-saturated 0.1 M KOH were carried out. As indicated by the linear sweep voltammetry (LSV) tests in Fig. 4a, Fe-SAs/NPS-HC exhibits superior ORR activity with a more positive half-wave ($E_{1/2}$) potential of 0.912 V than that of the state-of-art commercial Pt/C ($E_{1/2}$, 0.840 V). As shown in Fig. 4b, the kinetic current density ($J_k$) of 71.9 mA cm$^{-2}$ at 0.85 V for Fe-SAs/NPS-HC is a record value, to the best of our knowledge, and is higher than that of Pt/C (4.78 mA cm$^{-2}$) by a factor of 15. Strikingly, the better ORR kinetics of Fe-SAs/NPS-HC is further confirmed by a smaller Tafel slope of 36 mV dec$^{-1}$ compared with that of Pt/C (70 mV dec$^{-1}$). As listed in Supplementary Table 3, the ORR activity and kinetics of Fe-SAs/NPS-HC outperform most of the reported nonprecious catalysts, suggesting Fe-SAs/NPS-HC is one of the best ORR electrocatalysts to date. For comparison, the iron-free N, P, S co-doped hollow carbon polyhedron (NPS-HC) catalyst was synthesized (see details and Supplementary Figure 16) and exhibits significantly lower ORR performance. To investigate the effect of the hollow structure, we prepared N, P, S co-doped solid carbon polyhedron with isolated Fe atomic sites (Fe-SAs/NPS-C) catalyst (see details and Supplementary Figure 17). The as-prepared Fe-SAs/NPS-C exhibits apparently degraded ORR activity with a negatively shifted $E_{1/2}$ of 0.894 mV, which is 18 mV lower than Fe-SAs/NPS-HC (Fig. 4a). The slower ORR kinetics in Fe-SAs/NPS-C is revealed by smaller $J_k$ (34.6 mA cm$^{-2}$ at 0.85 V) and larger Tafel slope (51 mV dec$^{-1}$) (Fig. 4b, c). These results demonstrate the construction of hollow structure can efficiently improve ORR activity and accelerate ORR kinetics. As shown in Supplementary Figure 18, the electrochemical double-layer capacitance ($C_{dl}$) of Fe-SAs/NPS-HC

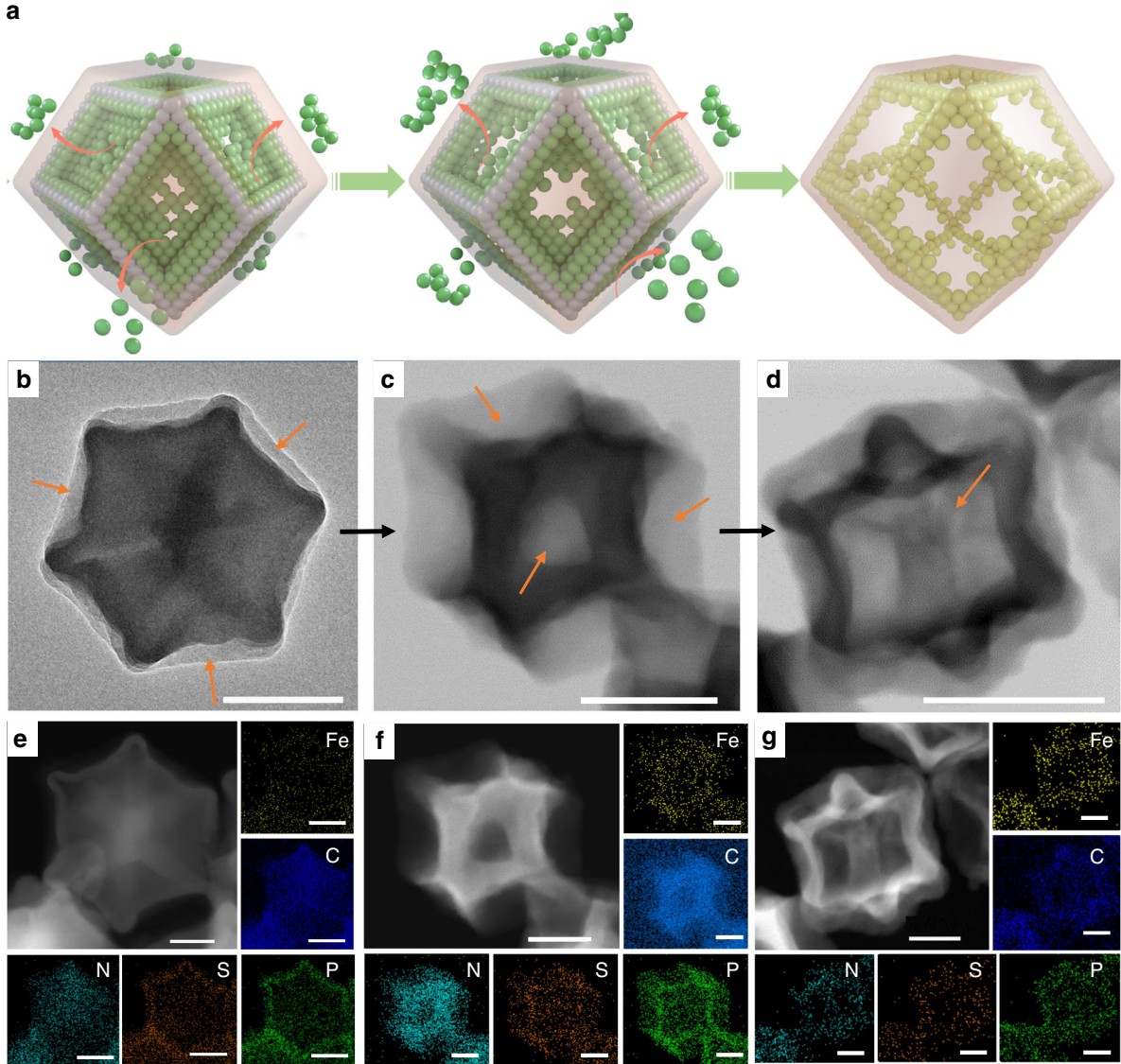

**Fig. 2** The formation process of functionalized hollow structure. **a** Schematic illustration of evolutionary trajectory of hollow structure. **b–d** Transmission electron microscopy (TEM) images, scale bar, 200 nm; and **e–g** the corresponding energy-dispersive spectroscopy (EDS) element maps (Fe: yellow, C: blue, N: cyan, S: orange, P: green) at different pyrolysis temperatures of 400, 500, and 600 °C for 30 min with a heating rate of 5 °C min$^{-1}$ under flowing Ar atmosphere, respectively, scale bar: 100 nm

(117.5 mF cm$^{-2}$) is higher than that of Fe-SAs/NPS-C (72.3 mF cm$^{-2}$) and NPS-HC (64.9 mF cm$^{-2}$), suggesting Fe-SAs/NPS-HC possesses a larger electrochemically active surface area (ESCA). The Nyquist plots of electrochemical impedance spectroscopy (EIS) demonstrate that Fe-SAs/NPS-HC exhibits a much smaller semicircle diameter than Pt/C and other reference catalysts, which represents a lower charge transfer resistance for Fe-SAs/NPS-HC, consistent with its more favorable charge transfer process (Fig. 4d).

To assess ORR pathway of Fe-SAs/NPS-HC, the RDE measurements at various rotation rates were carried out (Fig. 4e). The nearly parallel Koutecky−Levich (K−L) plots (inset of Fig. 4e) indicate the first-order reaction kinetics toward the concentration of dissolved oxygen. According to the K−L equation, the electron-transfer number ($n$) is calculated to be in the range 3.96−3.99 over the whole potential range, approaching the theoretical value of 4 for the 4e$^-$ ORR process (Supplementary Figure 19). Moreover, rotating ring disk electrode (RRDE) tests reveal H$_2$O$_2$ yield of Fe-SAs/NPS-HC remains below 4.2 %

in the potential range 0.3–0.9 V (Fig. 4f), corresponding to a high electron-transfer number of 3.90–4.00. For comparison, the $n$ value of 3.93–4.00 is achieved for the Pt/C. These results confirm Fe-SAs/NPS-HC undergoes a high-efficiency catalytic process via a 4e$^-$ ORR pathway. Unlike the Pt/C, Fe-SAs/NPS-HC exhibits outstanding tolerance to methanol crossover (Fig. 4g and Supplementary Figure 20). As shown in Fig. 4h, no obvious decay in $E_{1/2}$ is detected after 5000 cycles, indicating the excellent stability of Fe-SAs/NPS-HC. Furthermore, the detailed characterization of the used catalyst after durability test demonstrates atomic dispersion of iron atoms still remains (Supplementary Figures 21 to 24).

Fe-SAs/NPS-HC also exhibits outstanding ORR performance in acidic media (Supplementary Figure 25). By RDE measurements in 0.5 M H$_2$SO$_4$, an excellent ORR activity with an $E_{1/2}$ of 0.791 V is observed in Fe-SAs/NPS-HC, which approaches that of the commercial Pt/C (0.800 V), and is much better than those of Fe-SAs/NPS-C and NPS-HC (Supplementary Figure 25a,b). The favorable ORR kinetics of Fe-SAs/NPS-HC is verified by higher $J_k$

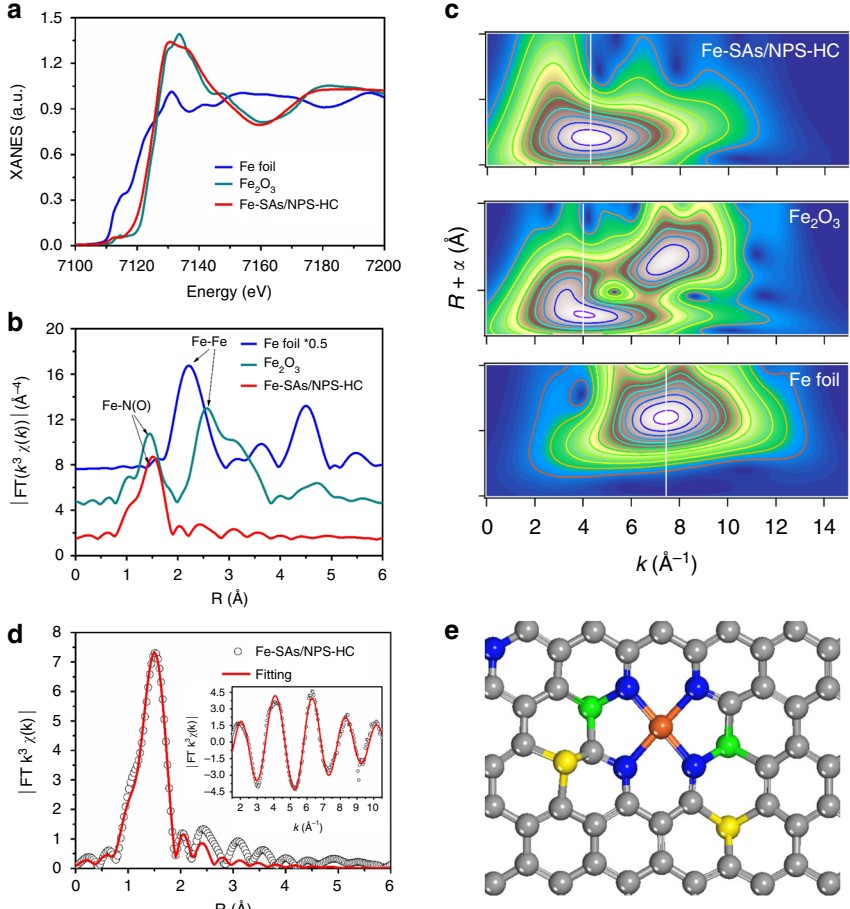

**Fig. 3** Atomic structural analysis. **a, b** Fe $K$-edge X-ray absorption near-edge structure (XANES) spectra (**a**) and Fe $K$-edge $k^3$-weighted Fourier transform (FT) spectra (**b**) of single iron atomic sites supported on a nitrogen, phosphorus and sulfur co-doped hollow carbon polyhedron (Fe-SAs/NPS-HC), $Fe_2O_3$ and Fe foil samples, respectively. **c** Wavelet transform (WT) of Fe-SAs/NPS-HC in comparison with $Fe_2O_3$ and Fe foil samples, respectively. **d** The corresponding extended X-ray absorption fine structure (EXAFS) $R$ space fitting curves; inset: The corresponding EXAFS $k$ space fitting curves of Fe-SAs/ NPS-HC. **e** Schematic model of Fe-SAs/NPS-HC, Fe (orange), N (blue), P (green), S (yellow) and C (gray)

of 18.8 mA cm$^{-2}$ and lower Tafel slope of 54 mV dec$^{-1}$ compared to Pt/C and Fe-SAs/NPS-C and NPS-HC (Supplementary Figure 25c). As shown in Supplementary Table 4, the ORR performance of Fe-SAs/NPS-HC surpasses that of most reported nonprecious metal ORR electrocatalysts. RRDE measurements demonstrate that Fe-SAs/NPS-HC exhibits a low $H_2O_2$ yield below 2.8% with a high electron-transfer number of 3.95, suggesting its high-efficiency 4e$^-$ ORR pathway (Supplementary Figure 25d and Supplementary Figure 26). The accelerated durability test reveals Fe-SAs/NPS-HC shows an outstanding stability with a little negative $E_{1/2}$ shift of 5 mV after 5000 cycles (Supplementary Figure 25e). Moreover, Fe-SAs/NPS-HC has a superior tolerance to carbon monoxide and methanol crossover compared to Pt/C (Supplementary Figure 25f and Supplementary Figure 27).

To investigate the effect of composition on ORR performance, iron-based single-atom catalysts with different content of Fe were prepared (Supplementary Figure 28 and Supplementary Table 5). As shown in Supplementary Figure 29, as the content of Fe increases, the corresponding ORR polarization curves gradually positively shifted and the values of $E_{1/2}$ gradually increase, suggesting enhanced ORR activity is attributed to the increase of the amount of Fe in catalysts. The rising $J_k$ and decreasing Tafel slope indicate that the crucial role of Fe component in accelerating ORR kinetics. The similar results are verified in acidic ORR tests. With the increase of the content of Fe, the ORR

activity and kinetics are gradually enhanced in term of better $E_{1/2}$, higher $J_k$ and smaller Tafel slope. These results suggest that Fe component as isolated atoms plays the key role in ORR performance in both alkaline and acidic media. To further reinforce the above conclusion, control experiments are designed and performed. It is known that SCN$^-$ ion has a strong affinity for Fe and can poison Fe−N$_4$ coordination in catalyzing ORR[42]. When 0.01 M KSCN is added in $O_2$-saturated 0.5 M $H_2SO_4$, the onset potential and half-wave potential exhibit obviously negative shifts, along with the visible decrease of diffuse-limited current (Supplementary Figure 30a). Then we rinsed this electrode with water and re-measured it in 0.1 M $O_2$-saturated KOH. It can be seen that ORR polarization curve gradually recovers to the pristine level as the LSV tests go on, which is ascribed to the recovery of the blocked Fe−N$_4$ sites owing to the dissociation of SCN$^-$ ion on the Fe active centers in 0.1 M KOH (Supplementary Figure 30b). These results unambiguously reveal that Fe component is essential to display outstanding ORR performance.

Moreover, we carried out in situ XAFS to investigate the electronic structure of active sites under ORR operating conditions. As shown in Fig. 4i, the Fe $K$-edge XANES curves of Fe-SAs/NPS-HC demonstrate the potential-dependent change in the oxidation states of Fe. The Fe $K$-edge XANES of in situ electrode above the onset potential at 1.0 V is similar to that of ex situ electrode, suggesting the excellent stability of active sites in electrolyte. With the applied potential from 1.0 to 0.3 V to

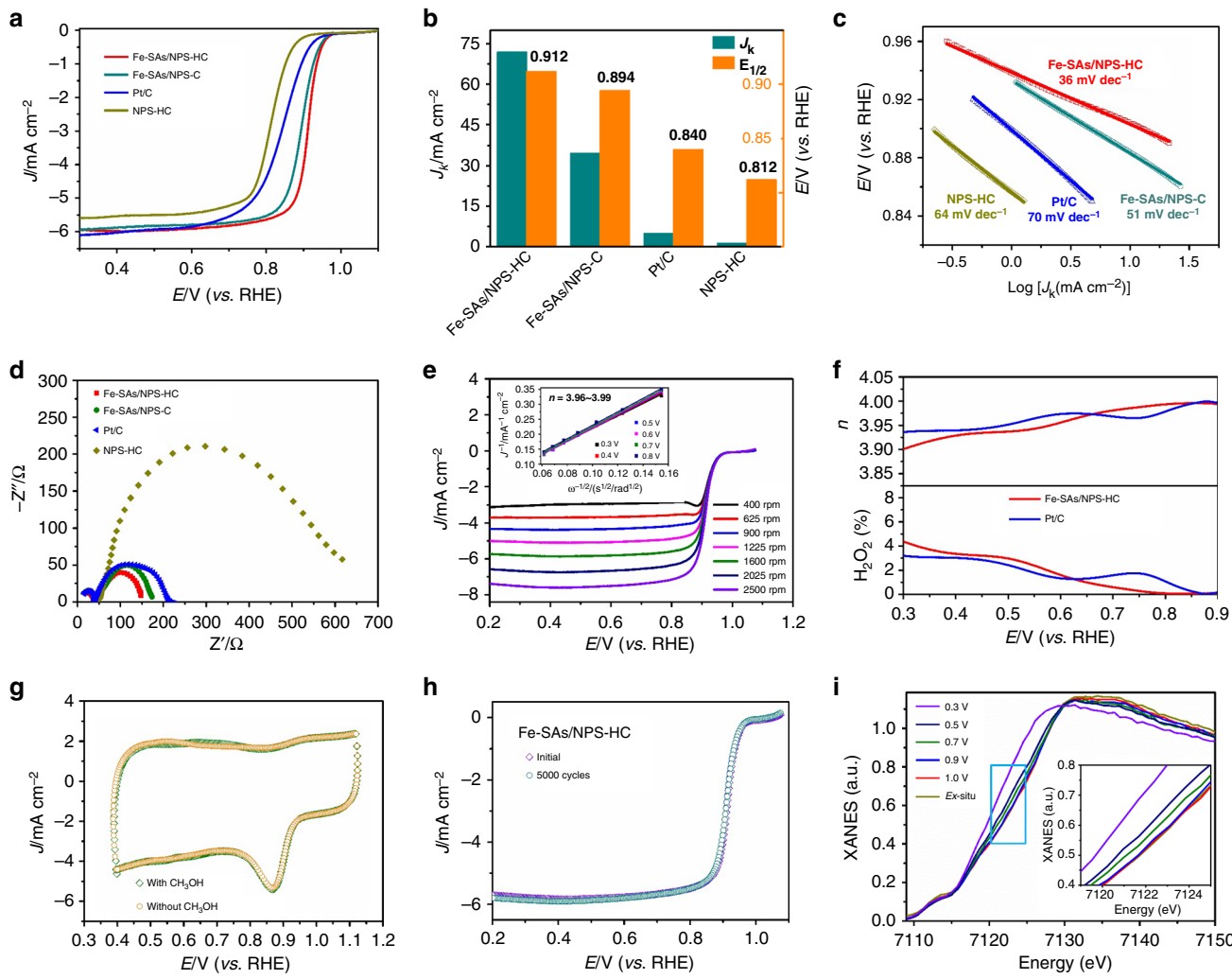

**Fig. 4** Electrocatalytic oxygen reduction reaction performance. **a** Oxygen reduction reaction (ORR) polarization curves for single iron atomic sites supported on a nitrogen, phosphorus and sulfur co-doped hollow carbon polyhedron (Fe-SAs/NPS-HC), N, P, S co-doped solid carbon polyhedron with isolated Fe atomic sites (Fe-SAs/NPS-C), iron-free N, P, S co-doped hollow carbon polyhedron (NPS-HC) and 20% Pt/C. **b** Comparison of $J_k$ at 0.85 V and $E_{1/2}$ of Fe-SAs/NPS-HC and the corresponding reference catalysts. **c** The Tafel plots for Fe-SAs/NPS-HC and the corresponding reference catalysts. **d** Nyquist plots of electrochemical impedance spectroscopy (EIS) over Fe-SAs/NPS-HC, Fe-SAs/NPS-C, NPS-HC and 20% Pt/C catalysts in 0.1 M KOH. **e** The ORR polarization curves at different rotating rates of Fe-SAs/NPS-HC (inset: K−L plots and electron-transfer numbers). **f** Electron-transfer number ($n$) (top) and $H_2O_2$ yield (bottom) vs. potential. **g** Cyclic voltammetry (CV) data for Fe-SAs/NPS-HC in $O_2$-saturated 0.1 M KOH without and with 1.0 M $CH_3OH$. **h** ORR polarization curves of Fe-SAs/NPS-HC before and after 5000 cycles. **i** Fe $K$-edge X-ray absorption near-edge structure (XANES) spectra of Fe-SAs/NPS-HC at various potentials during ORR catalysis in $O_2$-saturated 0.1 M KOH (inset: the enlarged Fe $K$-edge XANES spectra)

increase ORR current density, the Fe $K$-edge XANES shifts toward lower energy, indicating the oxidation state of Fe species gradually decreases. The in situ XAFS analysis clearly reveals that ORR performance closely correlates with the electronic structure of active sites.

**Electronic control effect**. To understand the nature of excellent ORR reactivity kinetics of Fe-SAs/NPS-HC, DFT calculations were carried out to investigate the free energetics of four-electron ORR reaction mechanism. To investigate the synergistic effects of nonmetal dopants, N-doped single-atom Fe sample (Fe-SAs/N-C), N, P co-doped single-atom Fe sample (Fe-SAs/NP-C), and N, P, S co-doped single-atom Fe sample (Fe-SAs/NPS-C) are considered; the free energy diagrams are presented in Fig. 5a. The optimized structures of different doped types of single-atomic-site Fe catalysts and the structures of the corresponding adsorbed intermediates are shown in Supplementary Figures 31 and 32. Among these three catalysts, all the intermediates have the

strongest binding energy on Fe-SAs/N-C, followed by Fe-SAs/ NP-C, while Fe-SAs/NPS-C has the weakest binding energy for the intermediates. On Fe-SAs/N-C, the last step is endothermic by 0.13 eV, indicating that too strong binding of OH* intermediate makes it not easy to be removed to produce OH⁻. On Fe-SAs/NP-C, the first three electron-transfer steps are exothermic, while the last step is almost thermo-neutral (−0.02 eV). All reaction steps are down-hilled in the 4e⁻ reduction path on Fe-SAs/NPS-C, suggesting ORR process is more thermodynamically favorable on Fe-SAs/NPS-C. As shown in Fig. 5a, when the electrode potential $U = 1.0$ V vs. RHE (i.e. $U = 0.23$ V vs. normal hydrogen electrode (NHE) at pH = 13) is applied on Fe-SAs/ NPS-C, all the elementary steps become thermo-neutral, except the second one (OOH* + OH⁻ + $H_2O(l)$ + 3e⁻ → O* + 2OH⁻ + $H_2O(l)$ + 2e⁻), which is exothermic by −0.91 eV. On the contrary, on both Fe-SAs/N-C and Fe-SAs/NP-C catalysts, the last two electrochemical steps become relatively strong endothermic (Supplementary Figure 33) at $U^{RHE} = 1.0$ V, therefore slowing

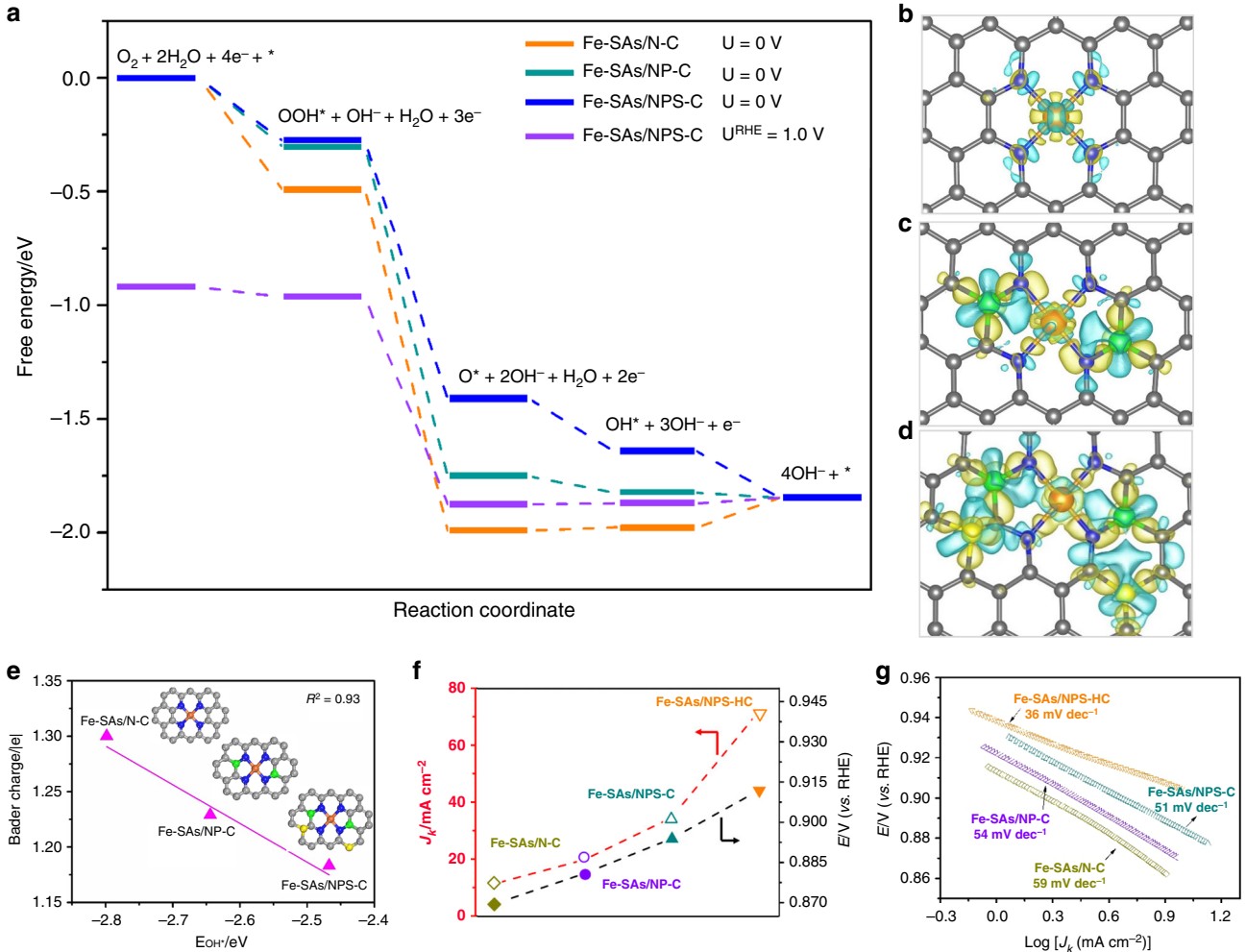

**Fig. 5** Electronic control effect. **a** Free energy diagram of the oxygen reduction reaction (ORR) on single-atom iron (Fe-SAs)/nitrogen-doped carbon (N-C), Fe-SAs/nitrogen, phosphorus co-doped carbon (NP-C) and Fe-SAs/nitrogen, phosphorus, sulfur co-doped carbon (NPS-C) ($U^{RHE} = U^{NHE} + 0.0591$ pH; $U^{RHE} = 0$ V, $U^{NHE} = -0.77$ V; $U^{RHE} = 1$ V, $U^{NHE} = 0.23$ V). **b–d** Calculated charge density differences of Fe-SAs/N-C (**b**), Fe-SAs/NP-C (**c**), and Fe-SAs/NPS-C (**d**). Yellow and blue areas represent charge density increase and reduction, respectively. The cutoff of the density-difference isosurfaces is equal to 0.01 electrons Å$^{-3}$. (C: gray, N: blue, Fe: orange, S: yellow, P: green). **e** Linear relationship between OH* binding energy and Bader charge of single-atom iron in Fe-SAs/N-C, Fe-SAs/NP-C and Fe-SAs/NPS-C, respectively. Insets: The corresponding schematic models of samples, Fe (orange), N (blue), P (green), S (yellow) and C (gray). **f, g** Comparison of $J_k$ at 0.85 V and $E_{1/2}$ (**f**) and the Tafel plots (**g**) for Fe-SAs/N-C, Fe-SAs/NP-C, Fe-SAs/NPS-C, and single iron atomic sites supported on a nitrogen, phosphorus and sulfur co-doped hollow carbon polyhedron (Fe-SAs/NPS-HC)

down the whole reaction kinetics. This result suggests that Fe-SAs/N-C and Fe-SAs/NP-C exhibit less reactivity and lower kinetics than Fe-SAs/NPS-C. According to the free energetics (Supplementary Tables 6–8), the rate-determining step on Fe-SAs/N-C and Fe-SAs/NP-C is the final electrochemical step, which is correlated with OH* binding energy. Therefore, a weakening of the OH* binding is expected to improve the ORR activity and kinetics[43]. The charge density differences (Fig. 5b–d) and Bader charge (Fig. 5e) analysis revealed that the electron donation from the surrounding sulfur and phosphorus can make the charge of metal center Fe (Fe$^{\delta+}$) of Fe-SAs/NPS-C become less positive, resulting in a less intense combination with OH*. As shown in Fig. 5e, it is well demonstrated that as the Bader charge of the single-atom Fe$^{\delta+}$ decreases, the OH* binding energy decreases linearly. Thus, Fe-SAs/NPS-C has a better "4e$^-$ reduction" catalytic performance and kinetics. The calculated free energy diagram on different doped types of Fe-SAs samples at 1.23 V (thermodynamic equilibrium potential) was presented in Supplementary Figure 34. The result is consistent with the results of computation at the $U^{RHE} = 0$ V and 1.0 V vs. RHE at the pH

= 13 condition. Based on the above analysis, the proposed reaction pathway has been demonstrated in Supplementary Figure 35. The catalytic properties of these three catalysts under acidic conditions are similar to that under alkaline conditions. At $U = 1.23$ V, the first two reduction steps on Fe-SAs/N-C are exothermic, and the last two steps are endothermic by 0.48 and 0.59 eV, respectively (Supplementary Figures 36 and 37). When P and S are doped, only the second step (OOH* + 3 H$^+$ + 3e$^-$ → O* + 2 H$^+$ + H$_2$O (l) + 2e$^-$) is exothermic, while the other three steps are endothermic. On Fe-SAs/NPS-C, the last two steps are endothermic by 0.23 and 0.27 eV, respectively, which are smaller than Fe-SAs/N-C and Fe-SAs/NP-C, indicating that the ORR process of Fe-SAs/NPS-C is thermodynamically favorable.

To further confirm that modulating the electronic properties of metal center by the doping of S, P atoms enables optimization of ORR kinetics and activity, we designed the control samples and evaluated their ORR performance. As shown in Fig. 5f and Supplementary Figure 38, the doping of heteroatoms leads to different degree of enhancement of ORR activity in view of $E_{1/2}$: N-doped Fe-SAs/N-C (0.870 V) < N, P co-doped Fe-SAs/NP-C

(0.881 V) < N, P, S co-doped Fe-SAs/NPS-C (0.894 V). Regarding the ORR kinetics, Fe-SAs/NPS-C exhibit higher $J_k$ of 34.6 mA cm$^{-2}$ at 0.85 V than those of Fe-SAs/NP-C (19.7 mA cm$^{-2}$) and Fe-SAs/N-C (11.7 mA cm$^{-2}$). The better ORR kinetics of Fe-SAs/NPS-C is further confirmed by smaller Tafel slope of 51 mV dec$^{-1}$ compared with Fe-SAs/NP-C (54 mV dec$^{-1}$) and Fe-SAs/N-C (59 mV dec$^{-1}$) (Fig. 5g). This strategy of electronic modulation of active center also applies to boost acidic ORR performance. As shown in Supplementary Figures 39 and 40, the N, P, S co-doped Fe-SAs/NPS-C exhibits obvious enhancement in activity with more positive $E_{1/2}$ of 0.764 V compared with N-doped Fe-SAs/N-C (0.666 V) and N, P co-doped Fe-SAs/NP-C (0.725 V) along with higher $J_k$ and better Tafel slope compared with Fe-SAs/N-C and Fe-SAs/NP-C. The above experimental observations are in agreement with the DFT results. It is worth mentioning that further strengthened ORR performance is observed over Fe-SAs/NPS-HC, which is attributed to its multiple advantages in synergistically improving ORR activity and kinetics. The larger electrochemically active surface area can expose more active sites and promote the adsorption and activation of ORR-relevant species on Fe-SAs/NPS-HC (Supplementary Figure 18). The lower charge transfer resistance is conducive to enhancing the efficiency of charge transfer (Supplementary Figure 41). The unique structure with atomically dispersed Fe−N$_4$ sites and electronic control effect from the surrounding S and P atoms contribute to improving the intrinsic activity of active sites, leading to enhanced reaction efficiency of ORR-relevant species on Fe-SAs/NPS-HC, as supported by its higher mass activity (MA) and turnover frequency (TOF) compared with other previously reported non-noble single-atom catalysts (Supplementary Table 9). The hollow structure functionalities with hierarchical porous of Fe-SAs/NPS-HC improve the accessibility of active sites and facilitate mass transport properties[3,44].

**Zinc-air battery and hydrogen-air fuel cell performance**. To assess the potential application of Fe-SAs/NPS-HC in energy storage and conversion devices, the Zn-air battery was assembled by applying Fe-SAs/NPS-HC as the air cathode and zinc foil as the anode with 6 M KOH electrolyte. For comparison, the Zn-air battery by using 20 wt% Pt/C as the air cathode was also made and tested in an identical condition. As shown in Fig. 6a, the Fe-SAs/NPS-HC-based battery exhibits a higher open circuit voltage of 1.45 V compared with the Pt/C-based battery, suggesting the Fe-SAs/NPS-HC-based battery can output higher voltage. The maximum power density of Fe-SAs/NPS-HC-based battery achieves as high as 195.0 mW cm$^{-2}$ with a high current density of 375 mA cm$^{-2}$ (Fig. 6b). Both power density and current density of Fe-SAs/NPS-HC-based battery outperforms that of Pt/C-based battery (177.7 mW cm$^{-2}$, 283 mA cm$^{-2}$). The rechargeability and cyclic durability of catalysts are of great significance for the practical applications of Zn-air battery. As shown in Fig. 6c, Fe-SAs/NPS-HC-based battery exhibits an initial charge potential of 2.07 V and a discharge potential of 1.11 V. Obviously, Fe-SAs/NPS-HC-based battery delivers a lower charge−discharge voltage gap compared with the Pt/C-based battery, indicating the better rechargeability of Fe-SAs/NPS-HC-based battery. Unlike the Pt/C-based battery with obvious voltage change, Fe-SAs/NPS-HC-based battery exhibits the negligible variation in voltage after 500 charging/discharging cycle tests with 200,000 s, suggesting the outstanding long-term durability.

Moreover, we exploited Fe-SAs/NPS-HC catalyst as a cathode to evaluate its performance in acidic proton exchange membrane fuel cells (PEMFCs). The Fe-SAs/NPS-HC-based membrane electrode assembly (MEA) was measured under realistic fuel cell operations with the cathode operated on air. Figure 6d shows the polarization and power density curves of PEMFCs at the testing temperature of 60 °C. Fe-SAs/NPS-HC-based MEA exhibits a remarkable current density of ~50 mA cm$^{-2}$ at 0.8 V, which is comparable to the highest current density of 75 mA cm$^{-2}$ reported to date[3]. The maximum power density of Fe-SAs/NPS-HC-based MEA is 333 mW cm$^{-2}$ at 0.41 V, which is ~92% the power density of the commercial Pt/C-based MEA under identical test conditions. When testing at 80 °C, Fe-SAs/NPS-HC-based MEA reaches a high power density of 400 mW cm$^{-2}$ at 0.40 V, approaching that of the commercial Pt/C-based MEA (Supplementary Figure 42). The fuel cells performance of Fe-SAs/NPS-HC-based MEA is compared favorably to that of most reported Pt-free catalysts (Supplementary Table 10), which is attributed to its unique structure functionalities with hierarchically porous property and highly active single Fe atomic sites. These results demonstrate that Fe-SAs/NPS-HC catalyst holds great promise in the application of PEMFC devices and Zn-air batteries. The mass production of catalysts is highly desirable for further commercial application. Indeed, our synthetic method is easily scaled up to grams of Fe-SAs/NPS-HC (Supplementary Figure 43). Furthermore, we extended our synthetic method to other metal single-atom catalysts and discovered the generality of our synthetic method (Supplementary Figures 44–47).

## Discussion

In summary, we develop a novel MOF@polymer strategy to fabricate an Fe-SAs/NPS-HC catalyst. Benefiting from a rational, optimized protocol via structure functionalities and electronic control of single iron atomic sites, Fe-SAs/NPS-HC accelerates sluggish ORR kinetics and achieves excellent ORR performance in both alkaline and acidic media. Moreover, Fe-SAs/NPS-HC exhibits great promise for Zn-air battery and H$_2$-air fuel cell devices. The vital role of the hollow structure for enhanced kinetics is disclosed by the experiments. DFT calculations reveal atomic dispersion of Fe coordinated by N atoms and an electronic effect from surrounding S and P atoms that contributes to high efficiency and satisfactory kinetics for ORR. This work provides a successful paradigm to strengthen catalytic kinetics and activity via structure functionalities and electronic control of active sites, which may be extended to the design and optimization of other catalysts with high efficiency.

## Methods

**Catalyst preparation**. Typically, for the synthesis of ZIF-8, Zn(NO$_3$)$_2$·6H$_2$O (2.380 g, 8 mmol) was dissolved in 60 mL mixture solvent of N,N-dimethylformamide (36 mL), ethyl alcohol (12 mL), and methanol (12 mL). Subsequently, 2-methylimidazole (2.627 g, 32 mmol), dissolved in the mixture solvent of N,N-dimethylformamide (12 mL) and methanol (8 mL), was added into the above solution with vigorous stirring for 24 h at room temperature. The as-obtained ZIF-8 product was centrifuged and washed with methanol for twice and finally dried at 80 °C for 12 h. Then, for the synthesis of ZIF-8/Fe@PZS, bis(4-hydroxyphenyl) sulfone (325 mg) and phosphonitrilic chloride trimer (152 mg) were dissolved in 100 mL of methanol, marked as solution A. Then, the as-obtained powder of ZIF-8 (400 mg) was dispersed in methanol (40 mL), followed by injecting 1.73 mL methanol solution of iron (III) nitrate nonahydrate (10 mg mL$^{-1}$). Subsequently, the dispersion and N,N-diethylethanamine (1 mL) were respectively added into the solution A, following by stirring for 15 h. The resulting precipitate marked as ZIF-8/Fe@PZS was collected, washed and finally dried in vacuum at 80 °C for 12 h. ZIF-8@PZS was prepared with the same synthesis procedure of ZIF-8/Fe@PZS except that iron (III) nitrate nonahydrate was not added. ZIF-8/Fe was prepared with the same synthesis procedure of ZIF-8/Fe@PZS except that 3.50 mL methanol solution of iron (III) nitrate nonahydrate (10 mg mL$^{-1}$) was used and solution A and N,N-diethylethanamine (1 mL) were not added. ZIF-8/Fe@PZM was prepared following the above synthesis procedure of ZIF-8/Fe@PZS except that bis(4-hydroxyphenyl) sulfone (325 mg) was replaced by bis(4-aminophenyl) ether (320 mg). The obtained powders of ZIF-8/Fe@PZS, ZIF-8@PZS, ZIF-8/Fe, and ZIF-8/Fe@PZM were placed in quartz boat, respectively, and then maintained at 900 °C for 3 h in a tube furnace with a heating rate of 5 °C min$^{-1}$ under flowing Ar atmosphere. When these samples cooled to room temperature, the as-prepared samples, Fe-SAs/NPS-HC, NPS-HC, Fe-SAs/N-C, and Fe-SAs/NP-C, were respectively collected and without any treatment for further use. In addition, for the synthesis of Fe-SAs/NPS-C, the obtained powder of Fe-SAs/NP-C (25 mg) was fully mixed with bis(4-

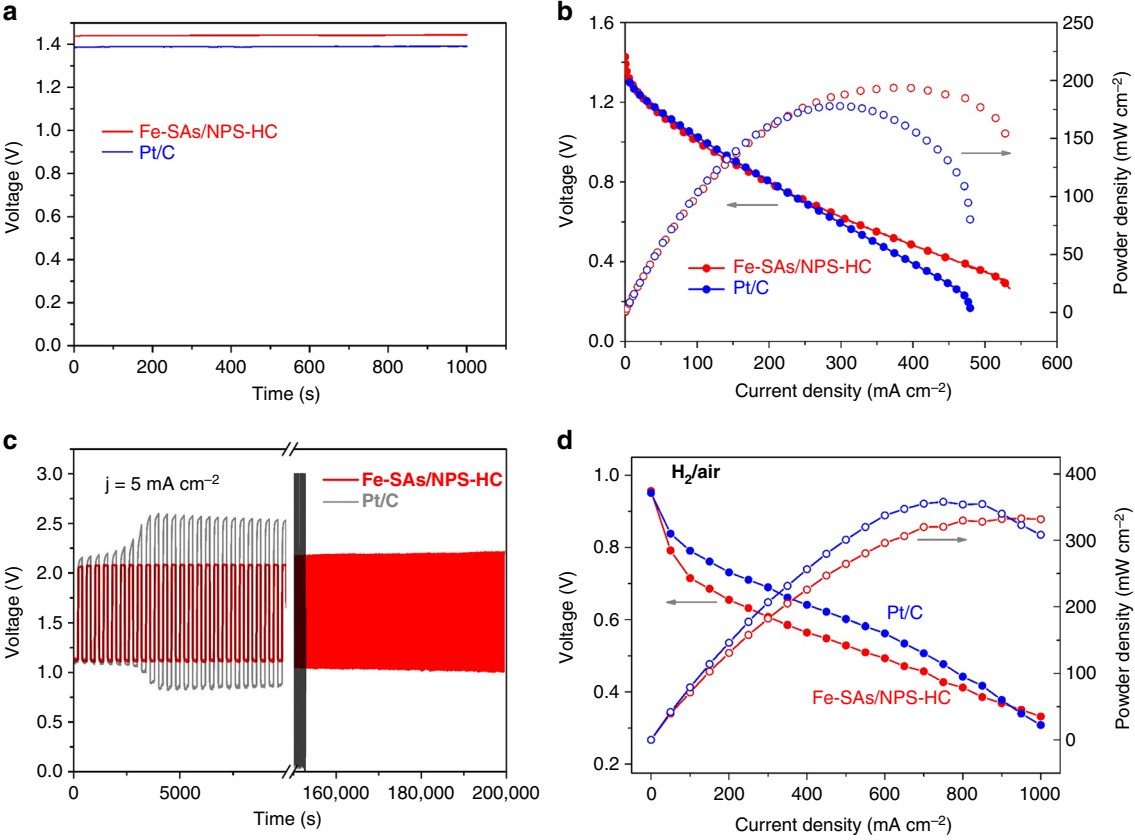

**Fig. 6** Zinc-air battery and hydrogen-air fuel cell performances. **a** The open circuit voltage curves of Zn-air batteries by applying single iron atomic sites supported on a nitrogen, phosphorus and sulfur co-doped hollow carbon polyhedron (Fe-SAs/NPS-HC) and 20 wt% Pt/C as the air cathode catalyst, respectively. **b** Discharging polarization curves and the corresponding power density plots of Fe-SAs/NPS-HC-based and 20 wt% Pt/C-based Zn-air batteries. **c** Charge−discharge cycling performance of Fe-SAs/NPS-HC-based and 20 wt% Pt/C-based Zn-air batteries. **d** H$_2$-air fuel cell polarization curves and power density plots of membrane electrode assemblies (MEAs) using Fe-SAs/NPS-HC (loading of 0.8 mg cm$^{-2}$) and 20 wt% Pt/C (loading of 0.8 mg cm$^{-2}$) as cathode catalysts, respectively. Membrane Nafion 211, cell 60 °C, electrode area 5 cm$^2$

hydroxyphenyl) sulfone (16 mg) and then placed in quartz boat and maintained at 800 °C for 2 h in a tube furnace under flowing Ar atmosphere, and the as-prepared sample was collected and without any treatment for further use. The synthesis of other Fe-SAs/NPS-HC samples with different Fe loadings are the same as that of Fe-SAs/NPS-HC except that the one-eighth of iron precursor, one-quarter of iron precursor, half of iron precursor, and double iron precursor are used, respectively, which are denoted as Fe-SAs/NPS-HC-1/8, Fe-SAs/NPS-HC-1/4, Fe-SAs/NPS-HC-1/2, and Fe-SAs/NPS-HC-2. Various M-SAs/NPS-HC catalysts (M = Ni, Cu, Ru, Ir) were synthesized through similar synthetic procedures, just by employing other metal precursors (Ni(NO$_3$)$_2$·6H$_2$O, Cu(NO$_3$)$_2$·3H$_2$O, RuCl$_3$·3H$_2$O and IrCl$_4$) to replace iron precursor.

**Characterization.** The crystalline structure and phase purity were assessed using PXRD (Rigaku RU-200b with Cu Kα radiation ($\lambda = 1.5406$ Å)). XPS experiments were performed on a ULVAC PHI Quantera microscope. The sizes and morphologies of samples were obtained on a Hitachi H-800 TEM. The high-resolution TEM and elemental mappings were collected on a JEOL JEM-2100F with electron acceleration energy of 200 kV. HAADF-STEM images were measured by using a JEOL 200F transmission electron microscope operated at 200 keV, equipped with a probe spherical aberration corrector. TGA was carried out using a Q5000. N$_2$ adsorption−desorption measurements were performed at 77 K on a Quantachrome SI-MP Instrument. The metal concentrations of the samples were measured by ICP-OES (Optima 7300 DV). $^{57}$Fe Mössbauer spectra were recorded at room temperature and at 78 K with a conventional constant-acceleration spectrometer using a $^{57}$Co(Rh) source. Doppler velocity and isomer shift (IS) were calibrated by metallic α-Fe foil. The XAFS spectra at Fe $K$-edge was acquired at 1W1B station in Beijing Synchrotron Radiation Facility (BSRF, operated at 2.5 GeV with a maximum current of 250 mA). The data of Fe-SAs/NPS-HC sample were recorded in fluorescence excitation mode using a Lytle detector. Fe foil and Fe$_2$O$_3$ were used as references and measured in a transmission mode using ionization chamber. The detailed ex situ XAFS and in situ XAFS analysis were given in Supplementary Information.

**Electrochemical measurement.** All the measurements were performed on a CHI 760E electrochemical station (CH Instruments, Inc., Shanghai) with a standard three-electrode system in 0.1 M KOH and 0.5 M H$_2$SO$_4$ electrolyte, respectively. To prepare a homogeneous dispersion, 5 mg catalyst samples (Fe-SAs/NPS-HC, NPS-HC) were dispersed in 1 mL of a mixture containing ethanol (0.495 mL), water (0.495 mL), and 0.5 wt% Nafion solution (10 μL), followed by sonicated for 1 h to form a homogeneous catalyst ink. For Fe-SAs/N-C, Fe-SAs/NP-C, and Fe-SAs/NPS-C, a certain amount of samples was prepared and dispersed in 1 mL of the above mixture solution to make the corresponding catalyst inks with the same Fe content as Fe-SAs/NPS-HC. One milligram of the commercial 20 wt% Pt/C was used to obtain catalyst ink. Then 20 μL of the catalyst suspension was pipetted onto a fresh glassy carbon (GC) electrode surface. A rotating disk electrode (RDE) with a GC disk (5 mm in diameter) and a rotating ring-disk electrode (RRDE) with a Pt ring (6.5 mm inner diameter and 8.5 mm outer diameter) and a GC disk of 5.5 mm diameter were used as the substrate for the working electrode. Ag/AgCl (saturated KCl solution) was used as reference and graphite rod as a counter electrode. The cyclic voltammetry (CV) tests were measured in N$_2$- and O$_2$-saturated 0.1 M KOH solution or 0.5 M H$_2$SO$_4$ with a sweep rate of 50 mV s$^{-1}$. RDE/RRDE tests were carried out in an O$_2$-saturated 0.1 M KOH solution or 0.5 M H$_2$SO$_4$ with O$_2$ purging with a scan rate of 10 mV s$^{-1}$ at different rotation rates. LSV polarization curves were obtained via RDE tests at a rotating speed of 1600 rpm. The ESCA of catalyst is estimated from electrochemical double-layer capacitance ($C_{dl}$), which is obtained by measuring double-layer charging from the CVs at different scan rates in non-Faradaic potential range in 0.1 M KOH. The linear slope of charging current vs. the scan rate represents $C_{dl}$. EIS was measured in the frequency range from 10$^6$ to 0.01 Hz. The long-term durability was measured by the accelerated durability test with continuously cycling between 0.6 and 1.0 V (vs. RHE) in O$_2$-saturated 0.1 M KOH or 0.5 M H$_2$SO$_4$. To test the tolerance to CO poisoning, chronoamperometric measurements at 0.45 V vs. RHE in a O$_2$-saturated mixed solution containing 0.5 M H$_2$SO$_4$ along with the injection of CO were performed. In this test process, CO gas (flow 50 mL s$^{-1}$) was purged through the GC surface at the time of 300 s and removed at the time of 600 s, followed by injection of O$_2$ gas (flow 100 mL s$^{-1}$) for the removal of CO gas. To test the tolerance to CH$_3$OH, chronoamperometric measurements at 0.45 V vs. RHE in a O$_2$-saturated mixed

solution containing 0.5 M $H_2SO_4$ along with the injection of $CH_3OH$ were performed. In this test process, $CH_3OH$ (5 mL) was added into the 0.5 M $H_2SO_4$ (125 mL) at the time of 250 s. All the potentials reported in this work were converted to the RHE. The detailed analysis was given in Supplementary Information.

**Hydrogen-air fuel cell measurements**. Membrane: Nafion211. MEA: 1 g catalysts were mixed with 10 mL deionized water under vigorous stirring. Then 75 mL Nafion 520 PFSA (5 wt% in isopropyl alcohol, EW~790, Dupont) solution was added to the mixture, followed by ultrasonic treatment for 30 min and a high-speed homogenizer (20,000 rpm) for 1 h to form catalyst slurry. Then the catalyst slurry was applied to PTFE thin film by spraying; after drying at 60 °C for 10 min followed with at 90 °C in $N_2$ atmosphere for 3 min, the catalyst layer was then transferred onto the membrane at 130 °C and 5 MPa by the decal method to form the catalyst-coated membrane (CCM). The GDL (Toray TGP-H-060) was placed on the anode and cathode side of the CCM to form the MEA. Catalyst loading: Sample #1, 0.8 mg cm$^{-2}$ (Pt) Pt/C (20 wt% Pt/C) in cathode and 0.15 mg cm$^{-2}$ (Pt) Pt/C (20 wt% Pt/C) in anode. Sample #2, 0.8 mg cm$^{-2}$ (total) the Fe-SAs/NPS-HC catalyst in cathode and 0.15 mg cm$^{-2}$ (Pt) Pt/C (20 wt% Pt/C) in anode. Single cell performance: The fuel cell performance was measured with G50 Fuel Cell Test Station (GreenLight Company) with back pressure of 0.2 MPa (absolute pressure), using $H_2$ as the fuel and air as the oxidant. The MEA was mounted in a single-cell test fixture with a serpentine flow field and a fuel cell clamp (with an active area of 25 cm$^2$). To control the RH of fuel gases, a water vapor saturated at a controlled dew point, was inflow slowly into gas inlet. After the cell activates to a stable value, polarization curves were recorded with the increase in current density regularly.

The assembled cell having active area of $5 \times 5$ cm$^2$ was operated at 60 °C or 80 °C under relative humidity of 100% for both anode and cathode. The gas stoichiometry was set to 1.5 for hydrogen in anode and 2.5 for air in cathode.

**Zinc-air battery measurements**. The Zn-air battery tests were performed in home-made electrochemical cells. The Fe-SAs/NPS-HC catalyst or the 20 wt% Pt/C catalyst was mixed with acetylene carbon black and polyvinylidene fluoride with the mass ratio of 8:1:1. Next, the mixed slurry was sprayed on the hydrophobic carbon paper uniformly as the air cathode. The mass loading of Fe-SAs/NPS-HC and the Pt/C was 1 mg cm$^{-2}$. A polished Zn foil was used as the anode electrode and 6 M KOH with 0.2 M zinc acetate dissolved was prepared as the electrolyte in the measurement.

**Computational details**. All theoretical calculations were performed using DFT, as implemented in the Vienna ab initio simulation package (VASP)[45,46]. The electron exchange and correlation energy was treated within the generalized gradient approximation in the Perdew−Burke−Ernzerhof functional (GGA-PBE)[47]. Iterative solutions of the Kohn–Sham equations were done using a plane-wave basis set defined by a kinetic energy cutoff of 400 eV. The $k$-point sampling was obtained from the Monkhorst−Pack scheme with a ($2 \times 2 \times 1$) mesh. The convergence criteria for the electronic self-consistent iteration and force were set to $10^{-4}$ eV and 0.03 eV Å$^{-1}$, respectively. The two-dimensional graphene support was modeled with a $6 \times 6$ supercell consisting of 72 carbon sites separated by a vacuum region of 15 Å along the direction normal to the sheet plane to avoid strong interactions between two adjacent layers. Two neighboring carbon atoms were removed to anchor an Fe atom, and four of carbon atoms were replaced by N atom, forming the Fe−$N_4$ site.

The adsorption energies were calculated according to the equation, $E_{ads} = E$ (adsorbate/substrate) $- [E(substrate) + E(adsorbate)]$, where $E$(adsorbate/substrate), $E$(substrate), and $E$(adsorbate) represent the total energy of substrate with adsorbed species, the clean substrate, and the molecule in the gas phase, respectively. The charge density differences were evaluated using the formula $\Delta\rho = \rho_{A+B} - \rho_A - \rho_B$, where $\rho_X$ is the electron density of X. Atomic charges were computed using the atom-in-molecule (AIM) scheme proposed by Bader[48,49].

The change in Gibbs free energy ($\Delta G$) for all the ORR steps are calculated based on the computational hydrogen electrode method developed by Nørskov et al.[50]. At standard condition ($U = 0$, pH 0, $p = 1$ bar, $T = 298$ K), the free energy $\Delta G_0$ is defined as $\Delta G_0 = \Delta E + \Delta ZPE + \Delta_0 \rightarrow_{298K} \Delta H - T\Delta S$, where $\Delta E$ is the energy change obtained from DFT calculations; $\Delta ZPE$, $\Delta H$, and $\Delta S$ denote the difference in zero point energy, enthalpy, and entropy due to the reaction, respectively. The enthalpy and entropy of the ideal gas molecule were taken from the standard thermodynamic tables[51]. Therefore, reaction free energy is further calculated by the equation: $\Delta G$ ($U$, pH, $p = 1$ bar, $T = 298$ K) $= \Delta G_0 + \Delta G_{pH} + \Delta G_U$, where $\Delta G_{pH}$ is the correction of the free energy of H$^+$-ions at a pH different from 0: $\Delta G_{pH} = -kT\ln[H^+] = kT\ln10 \times$ pH. $\Delta G_U = -neU$, where $U$ is the applied electrode potential, $n$ is the number of electrons transferred. Hence, the equilibrium potential $U^0$ for ORR at pH = 13 was determined to be 0.462 V (vs. NHE). The free energy of $O_2(g)$ was derived as $G_{O2(g)} = 2G_{H2O(l)} - 2G_{H2} + 4.92$ eV, and the free energy of OH$^-$ was calculated by $G_{OH-} = G_{H2O(l)} - G_{H+}$. The overall reaction of $O_2$ reduction to OH$^-$ in alkaline environment is: $O_2 + 2H_2O + 4e^- \rightarrow 4OH^-$, which is divided into the following steps[4,52]:

(1)  $O_2\,(g) + 2H_2O\,(l) + 4e^- + * \rightarrow OOH^* + OH^- + H_2O\,(l) + 3e^-$

(2)  $OOH^* + OH^- + H_2O\,(l) + 3e^- \rightarrow O^* + 2OH^- + H_2O\,(l) + 2e^-$

(3)  $O^* + 2OH^- + H_2O\,(l) + 2e^- \rightarrow OH^* + 3OH^- + e^-$

(4)  $OH^* + 3OH^- + e^- \rightarrow 4OH^- + *$

 $*$ indicates the adsorption site.

## Data availability

The data that support the findings of this study are available from the corresponding author upon request.

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

## Acknowledgements
This work was supported by China Ministry of Science and Technology under Contract of 2016YFA (0202801), and the National Natural Science Foundation of China (21521091, 21390393, U1463202, 21471089, 21671117). We thank Beijing Light Source for use of the instruments.

## Author contributions
Y.C. performed the experiments, collected, analyzed the data and wrote the manuscript. S.J. carried out the characterizations of samples, analyzed the data and wrote the manuscript. S.Z. contributed to the computational results and helped to revise the paper. W.C. and J.D. performed XAFS data analysis. W.-C.C. helped to test and analyzed HAADF-STEM results. R.S. and Z.Z. helped to test the catalytic performance and analysis. X.W. provided the resource for DFT calculations. L.Z. helped to perform XAFS measurements of samples. A.I.R. helped to test the Mössbauer spectroscopy measurements and analyze data and also helped to revise the manuscript. S.C. and H.T. helped to test the Zn-air battery and $H_2$/air fuel cell performance and analyze data. C.C. and Q.P. helped to analyze the data and contributed to the revising of the manuscript. D.W. and Y.L. conceived the experiments, planned synthesis, analyzed results, designed the research project and wrote the manuscript. All the authors commented on the manuscript and have given approval to the final version of the manuscript.

## Additional information

**Competing interests:** The authors declare no competing interests.

