## [Peer Review File · Nature Communications]

Reviewers' Comments:

Reviewer #1:

Remarks to the Author:

Despite the fact that article entitled "Structure functionalities and electronic control of single iron atomic sites catalyst for enhanced oxygen reduction" deals with very important for whole community problem of improving the ORR PGM-free catalysts the study performed cannot be considered for publication in Nature Communications due to the lack of scientific novelty and absence of outstanding results.

The utilization of MOFs as well as polymers for high temperature synthesis of PGM-free ORR catalysts well know. The performance of all M-N-C type of materials in alkaline media superior to Pt/C (common knowledge). The absence of state-of-the-art MEA data with presented ORR catalyst does not allow the researchers in the field to get any new information, previously not presented in open literature.

Decision – Decline.

Reviewer #2:

Remarks to the Author:

Chen et al. report in this manuscript a MOF@polymer strategy to synthesize single Fe atomic sites catalysts for the ORR. It's an interesting piece of work that integrates vogue concepts such as single atom catalysis, hollow nanostructure, electronic modulation, and DFT calculation. Surprisingly, the electrocatalytic performance of the titled material is striking. A half-wave potential of 0.912 V and a Tafel slope of 36 mV/dec are unprecedented, so far as I know. Besides, the materials are clearly characterized and the manuscript is well organized. Thus, I'd recommend the publication of this manuscript if the authors could properly address the following specific comments.

1, To me, the record performance of the synthesized catalyst is very exciting. Although DFT studies are provided to understand the results, more electrochemical tests are required, together with mechanistic discussions. Why Fe-SAs/NPS-HC shows an apparent lower Tafel slope than both Fe-SAs/NPS-C and Pt/C? Is it indicative of a different ORR pathway? How about the ECSA of different catalyst samples? Hollow structures are expected to favor mass transportation and charge transfer. Any evidence to support the superiority?

2, The stability of single atomic Fe catalysts is impressive. Any agglomeration or leaching of the single sites? Any compositional or shape evolution of the catalyst upon cycling? Characterization of the catalysts after cycling tests is necessary.

3, Looking at the Fe K-edge XANES curve, the profiles between Fe₂O₃ and the single atom sample are very close, particularly in the pre-peak region. This similarity implies that Fe is likely in Fe(III) valence. If so, the superior activity is out of expectation since Fe(III) species are believed to be ORR inactive compared to Fe(0). Also, fitting of the XANES spectra is quite bad. The authors are suggested to perform other analysis of the Fe state such as the useful Mossbauer spectroscopy.

4, In DFT calculation of the OER process, why U=0 V and 1.0 V are selected? I'd suggest computation at the thermodynamic equilibrium potential of ORR, i.e., U=1.23V.

5, Applying K-L equation to determine electron transfer number in ORR sometimes gives rise to abnormal value. I would recommend including a comparison of Pt disk electrode data. RRDE tests are useful to calibrate n values as well.

Reviewer #3:

Remarks to the Author:

The authors reported a N, P, S co-doped atomic Fe based carbon catalyst for efficient oxygen

reduction reaction. The doped P and S are proposed to enhance the activity of Fe-N-C configuration. This article provide a new sight that the multi dopants could improve the performance of atomic metal species. The manuscript is well written and recommended to be published on Nature Communications after several issues being addressed.

1. Atomic metal based catalysts were reported to be active for HER, OER and ORR, especially for the four coordinated atomic metals. Two important publication in this field should be cited (CHEM, 2018, 4, 285; Mater. Chem. Front., 2018, 2, DOI: 10.1039/c8qm00070k).
2. The authors should give the content of each element in the sample Fe-SAs/NPS-HC, since from the XPS spectra of the C 1s, the fractions of C-P, C-S and C-N are comparable to C=C indicating a large amount of heteroatom dopants. The content of element is an important information to assist further constructing the models for DFT calculations.
3. From the XPS spectrums of P 2p and N 1s, oxygen is confirmed in the sample. Does oxygen have any effects to the final ORR performance (being involved or accelerating the reaction)?
4. The authors verified the positive effect to the ORR performance of the hollow structure by comparing the performances of Fe-SAs/NPS-HC and the corresponding bulk sample. However, is this bulk sample has the similar component (amount of each element) with Fe-SAs/NPS-HC?
5. Please calculate the mass activity and turnover frequency of Fe-SAs/NPS-HC and compare with other atomic metal based catalysts. Normally, atomic metal based catalysts possess large mass activities and turnover frequencies.
6. The authors constructed a DFT model of Fe-SAs/NPS-HC. How to identify this model in the sample? Is there any direct or indirect evidence to guide setting up this model?
7. The Fe-N coordination structure is deduced from the XAS characterization. However, do Fe-S and Fe-P coordination structures also exist in the sample?
8. Since the catalyst Fe-SAs/NPS-HC has a remarkable ORR activity in the alkaline media, did the authors try it in the acidic media?

Response to Reviewer #1 (Remarks to the Author):

Comments: Despite the fact that article entitled "Structure functionalities and electronic control of single iron atomic sites catalyst for enhanced oxygen reduction" deals with very important for whole community problem of improving the ORR PGM-free catalysts the study performed cannot be considered for publication in Nature Communications due to the lack of scientific novelty and absence of outstanding results.

The utilization of MOFs as well as polymers for high temperature synthesis of PGM-free ORR catalysts well know. The performance of all M-N-C type of materials in alkaline media superior to Pt/C (common knowledge). The absence of state-of-the-art MEA data with presented ORR catalyst does not allow the researchers in the field to get any new information, previously not presented in open literature.

Decision – Decline.

Response: We thanks the reviewer and appreciate the suggestion, which has helped us much in improving the quality of our manuscript. In compliance with your comment and suggestion, we have employed our Fe-SAs/NPS-HC catalyst as a cathode to evaluate its performance in acidic proton exchange membrane fuel cells (PEMFCs). The Fe-SAs/NPS-HC based membrane electrode assembly (MEA) was measured under realistic fuel cell operations with the cathode operated on air.

Firstly, steady-state rotating disk electrode (RDE) measurements in 0.5 M H₂SO₄ solution were carried out to assess the ORR performance of Fe-SAs/NPS-HC in the acidic media (**Supplementary Fig. 25**). As revealed by the steady-state RDE polarization plots, an excellent ORR activity with a $E_{1/2}$ of 0.791 V is observed in Fe-SAs/NPS-HC, which approaches that of the

commercial Pt/C (0.800 V), and is much better than those of Fe-SAs/NPS-C and NPS-HC. The favorable ORR kinetics of Fe-SAs/NPS-HC is verified by higher kinetic current density of 18.8 mA cm⁻² and lower Tafel slope of 54 mV dec⁻¹ compared to Pt/C and Fe-SAs/NPS-C and NPS-HC. As shown in **Supplementary Table 3**, the ORR performance of Fe-SAs/NPS-HC surpasses that of most reported non-precious metal ORR electrocatalysts. RRDE measurements demonstrate that Fe-SAs/NPS-HC exhibits a low H₂O₂ yield below 2.8% with a high electron transfer number of 3.95, suggesting its high-efficiency 4e⁻ ORR pathway. The accelerated durability test reveals Fe-SAs/NPS-HC shows an outstanding stability with a little negative $E_{1/2}$ shift of 5 mV after 5000 cycles.

- We have added this supplementary discussion in the revised manuscript (paragraph beginning in Page 12, Line 29, highlighted in **BLUE**). And the corresponding Supplementary Fig. 25, Supplementary Fig. 26, Supplementary Fig. 27 and Supplementary Table 3 have been added in the revised Supplementary Information.

Supplementary Figure 25 | Electrocatalytic ORR performance of Fe-SAs/NPS-HC in 0.5 M

H₂SO₄ solution. (a) ORR polarization curves for Fe-SAs/NPS-HC, Fe-SAs/NPS-C, NPS-HC and 20% Pt/C. (b) Comparison of J_k at 0.75 V and $E_{1/2}$ of Fe-SAs/NPS-HC and the corresponding reference catalysts. (c) The Tafel plots for Fe-SAs/NPS-HC and the corresponding reference catalysts. (d) Electron transfer number (n) (top) and H₂O₂ yield (bottom) versus potential. (e) ORR polarization curves of Fe-SAs/NPS-HC before and after 5000 potential cycles. (f) Tolerance to CO poison of Fe-SAs/NPS-HC compared with 20% Pt/C at 0.45 V (vs. RHE). CO (flow 50 mL·s⁻¹) is injected into the 0.5 M H₂SO₄ solution at the time of 300 s and stopped at the time of 600 s.

Supplementary Table 4 | Comparison of performance for Fe-SAs/NPS-HC and other non-noble catalysts reported in the literatures in 0.5 M H₂SO₄.

Electrocatalysts	Rotation and scan rate	$E_{1/2}$ (V vs. RHE)	current density (mA/cm ²) at 0.75 V	Kinetic Current Density (mA/cm ²) 0.75 V	Limited current	Ref.
Fe-SAs/NPS-HC	900 rpm and 10 mVs⁻¹	0.801	3.21	21.9	3.76	Our work
	1600 rpm and 10 mV s⁻¹	0.791	3.95	18.8	5.01	
PANI-FeCo-C	900 rpm and 10 mVs ⁻¹	0.805	2.9	11.3	3.9	Science 2011 , 332, 443.
N-Fe-MOF	900 rpm and 10 mVs ⁻¹	0.79	2.4	6.67	3.75	Adv. Mater. 2014 , 26, 1378
Fe-CNT-PA1600	1600rpm and 5 mV s ⁻¹	0.65	1.1	1.35	5.9	Energy Environ. Sci. 2015 , 8, 1799.
Co-N-GA	1600rpm and 10 mV s ⁻¹	0.73	2.5	4.39	5.8	ACS Appl. Mater. Interfaces 2016 , 8, 6488.
Fe-N-CC	1600 rpm and 10 mV s ⁻¹	0.62	0.5	0.583	3.5	ACS Nano 2016 , 10, 5922.
ISAS-Co/HNCS	1600rpm and 10 mV s ⁻¹	0.773	3.9	12.8	5.61	J. Am. Chem. Soc. 2017 , 139, 17269.
(CM+PANI)-Fe-C	900 rpm and 10 mVs ⁻¹	0.8	2.6	19.5	3	Science 2017 , 357, 479

SA-Fe/NG	1600 rpm and 5 mV s ⁻¹	0.8	3.9	17.7	5	PNAS. 2018 , 115, 6626
----------	---	-----	-----	------	---	---

The H₂/air PEMFC performance was evaluated by using Fe-SAs/NPS-HC as a cathode catalyst layers in MEA under realistic fuel cell operations with the cathode operated on air. **Figure 6d** shows the polarization and power density curves of PEMFCs at the testing temperature of 60 °C. Fe-SAs/NPS-HC based MEA exhibits a remarkable current density of ~ 50 mA cm⁻² at 0.8 V, which is comparable to the highest current density of 75 mA cm⁻² reported to date (please refer to *Science* 2007, 357, 479). The maximum power density of Fe-SAs/NPS-HC based MEA is 333 mW cm⁻² at 0.41 V, which is ~ 92 % the power density of the commercial Pt/C based MEA under identical test conditions. When testing at 80 °C, Fe-SAs/NPS-HC based MEA reach a high power density of 400 mW cm⁻² at 0.40 V, approaching that of the commercial Pt/C based MEA (**Supplementary Figure 34**). The fuel cells performance of Fe-SAs/NPS-HC based MEA is compared favorably to that of most reported Pt-free catalysts (**Supplementary Table 9**), which is attributed to its unique structure functionalities with hierarchically porous property and highly active single Fe atomic sites. Moreover, unlike the commercial Pt/C catalyst with obvious decrease of catalytic activity, Fe-SAs/NPS-HC shows a superior tolerance to methanol crossover and carbon monoxide (**Supplementary Fig. 25f and Supplementary Figure 27**), which is beneficial to its commercial application in fuel cell devices. These results demonstrate that Fe-SAs/NPS-HC exhibits great potential in the substitution of the expensive Pt-based electrocatalysts to drive the cathodic ORR in PEMFC devices.

- The discussion about the H₂/air PEMFC performance of Fe-SAs/NPS-HC has been added in the revised manuscript (paragraph beginning in Page 18, Line 21, highlighted in **BLUE**). The corresponding Fig. 6 has been added in the revised manuscript. The corresponding test procedure has been described in the section of Methods in the revised manuscript (paragraph beginning in Page 23, Line 11, highlighted in **BLUE**). And the corresponding Supplementary Fig. 25f, Supplementary Fig. 27, Supplementary Fig. 33 and Supplementary Table 9 have been added in the revised Supplementary Information.

Figure 6 | Zn-air battery and H₂/air fuel cell performances. (a) The open circuit voltage curves of Zn-air batteries by applying Fe-SAs/NPS-HC and 20 wt% Pt/C as the air cathode catalyst, respectively. (b) Discharging polarization curves and the corresponding power density plots of Fe-SAs/NPS-HC based and 20 wt% Pt/C based Zn-air batteries. (c) Charge-discharge cycling performance of Fe-SAs/NPS-HC based and 20 wt% Pt/C based Zn-air batteries. (d) H₂/air fuel cell polarization curves and power density plots of MEAs using Fe-SAs/NPS-HC (loading of 0.8 mg cm⁻²) and 20 wt% Pt/C (loading of 0.8 mg cm⁻²) as cathode catalysts, respectively. Membrane Nafion 211, cell 60 °C, electrode area 5 cm².

Supplementary Figure 34 | H₂/air fuel cell polarization curves and power density plots of MEAs using Fe-SAs/NPS-HC (loading of 0.8 mg cm⁻²) and 20 wt% Pt/C (loading of 0.8 mg cm⁻²) as cathode catalysts, respectively. Membrane Nafion 211, cell 80 °C, electrode area 5 cm².

Supplementary Table 9 | Comparison of H₂/air fuel cell performance by using Fe-SAs/NPS-HC and other non-noble catalysts reported in the literatures as the cathode

Catalysts	Catalyst loading (mg/cm ²)	Back pressure (MPa)	Operation temperature (°C)	Peak power density (mW/cm ²)	Ref.
Fe-SAs/NPS-HC	0.8	0.2	60	333	Our work
		0.2	80	400	
(CM+PANI)-Fe-C	4	0.1	80	420	Science 2017 , 357, 479-484.
Fe-N-C-Phen-PANI	4	0.138	80	380	Adv. Mater. , 2017 ,29,1604456
Fe-P-C_Ar-NH900	4	0.138	80	335	Nano Energy 2016 , 26, 267-275.
Fe-MBZ	3.0 ± 0.5	0.4	80	330	J. Power Sources 2016 , 326, 43-49.
FePhen@MOF-ArNH ₃	3	0.25	80	380	Nat. Commun. 2015 , 6, 7343.
Fe/oPD-Mela	3	0.20	80	270	Appl. Catal., B 2014 , 158-159, 60-69.
Fe/PI-1000-III-NH ₃	4	0.2	80	320	J. Mater. Chem. A 2014 , 2, 11561-11564.
Fe/TPTZ/ZIF-8	1.14	0.1	80	300	Angew. Chem. Int. Ed. 2013 , 125, 7005-7008
Co-PPY-C	0.06 (Co loading)	0.2	80	70	Nature 2006 , 443, 63-66.

In addition, in order to extend the potential application of Fe-SAs/NPS-HC in energy storage and conversion devices, the Zn-air battery was assembled by applying Fe-SAs/NPS-HC as the air cathode and zinc foil as the anode with 6 M KOH electrolyte. For comparison, the Zn-air battery by using 20 wt% Pt/C as the air cathode was also made and tested in an identical condition. As shown in **Figure 6a**, the Fe-SAs/NPS-HC based battery exhibits a higher open circuit voltage of 1.45 V compared with the Pt/C based battery, suggesting the Fe-SAs/NPS-HC based battery can output higher voltage. The maximum power density of Fe-SAs/NPS-HC based battery achieves as

high as 195.0 mW cm^{-2} with a high current density of 375 mA cm^{-2} (**Figure 6b**). Both power density and current density of Fe-SAs/NPS-HC based battery outperforms that of Pt/C based battery (177.7 mW cm^{-2} , 283 mA cm^{-2}). The rechargeability and cyclic durability of catalysts are of great significance for the practical applications of Zn-air battery. As shown in **Figure 6c**, Fe-SAs/NPS-HC based battery exhibits an initial charge potential of 2.07 V and a discharge potential of 1.11 V. Obviously, Fe-SAs/NPS-HC based battery delivers a lower charge-discharge voltage gap compared with the Pt/C based battery, indicating the better rechargeability of Fe-SAs/NPS-HC based battery. Unlike the Pt/C based battery with obvious voltage change, Fe-SAs/NPS-HC based battery exhibits the negligible variation in voltage after 500 charging/discharging cycle tests with 200000 s, suggesting the outstanding long-term durability.

- The discussion about the Zn-air battery performance of Fe-SAs/NPS-HC has been added in the revised manuscript (paragraph beginning in Page 18, Line 3, highlighted in **BLUE**). The corresponding Fig. 6 has been added in the revised manuscript. The corresponding test procedure has been described in the section of Methods in the revised manuscript (paragraph beginning in Page 24, Line 1, highlighted in **BLUE**).

Moreover, we carried out *in situ* XAFS to investigate the electronic structure of active sites under ORR operating conditions. As shown in **Figure 4i**, the Fe K-edge XANES curves of the Fe-SAs/NPS-HC catalyst demonstrate the potential-dependent change in the oxidation states of Fe. The Fe K-edge XANES of *in situ* electrode above the onset potential such as at 1.0 V is similar to that of *ex situ* electrode, suggesting the excellent stability of active sites in electrolyte. With the applied potential from 1.0 V to 0.3 V to increase the ORR current density, the Fe K-edge XANES shifts toward lower energy, which indicates that the oxidation state of Fe species gradually decreases. The *in situ* XAFS analysis clearly reveals that the ORR performance closely correlates with the electronic structure of active sites. This result is consistent with our DFT calculations (**Figure 5**). The charge density differences (**Figure 5b-d**) and Bader charge (**Figure 5e**) analysis demonstrated that the electron donation from the surrounding sulfur and phosphorus can make the charge of metal center Fe ($\text{Fe}^{\delta+}$) of Fe-SAs/NPS-C become less positive, which contributes to a weakened binding of the adsorbed OH species, resulting in the enhanced ORR kinetics and efficiency. And we designed the control catalysts and assessed their ORR performance to further confirm that modulating the electronic properties of metal center by the surrounding S, P atoms can promote ORR kinetics and activity (**Figure 5f and 5g**). *In situ* XAFS analysis in conjunction with designed experiments as well as theoretical calculations verified that electronic control of active sites at the atomic scale is an effective strategy to enhance ORR performance.

- The discussion about *in situ* XAFS analysis has been added in the revised manuscript (paragraph beginning in Page 13, Line 15, highlighted in **BLUE**). The corresponding Fig. 4i has been added in the revised manuscript. The corresponding test procedure has been described in the revised Supplementary Information (paragraph beginning in Page S2, Line 18, highlighted in **BLUE**).

Generally speaking, in this work, we not only developed a novel and simple MOF@polymer strategy to simultaneously achieve the synthesis of unique atomically dispersed active sites, the construction of functionalized hollow structure via Kirkendall effect and electronic modulation of active centers by near-range coordination with nitrogen and long-range interaction with sulfur and

phosphorus, but also fabricated a remarkable ORR electrocatalyst (Fe-SAs/NPS-HC) with excellent performance in both alkaline and acidic media. In addition, the new concept of material design via structure functionalities and electronic control of active sites at the atomic scale are developed and evidenced. We believe that this design concept can offer an important guideline for researchers to design and optimize catalysts for achieving highly efficient catalytic process. Moreover, our catalyst exhibits a great potential application in PEMFCs devices and Zn/air battery. We think that our work will attract extensive research attention and promote the development of nonprecious metal catalysts in the application of energy conversion and storage devices. We hope that you can agree to publish our work in *Nature Communications* after modification. Your consideration of our work for publication in *Nature Communications* will be appreciated very much.

Response to Reviewer #2 (Remarks to the Author):

Chen et al. report in this manuscript a MOF@polymer strategy to synthesize single Fe atomic sites catalysts for the ORR. It's an interesting piece of work that integrates vogue concepts such as single atom catalysis, hollow nanostructure, electronic modulation, and DFT calculation. Surprisingly, the electrocatalytic performance of the titled material is striking. A half-wave potential of 0.912 V and a Tafel slope of 36 mV/dec are unprecedented, so far as I know. Besides, the materials are clearly characterized and the manuscript is well organized. Thus, I'd recommend the publication of this manuscript if the authors could properly address the following specific comments.

Comment 1: To me, the record performance of the synthesized catalyst is very exciting. Although DFT studies are provided to understand the results, more electrochemical tests are required, together with mechanistic discussions. Why Fe-SAs/NPS-HC shows an apparent lower Tafel slope than both Fe-SAs/NPS-C and Pt/C? Is it indicative of a different ORR pathway? How about the ECSA of different catalyst samples? Hollow structures are expected to favor mass transportation and charge transfer. Any evidence to support the superiority?

Response: We appreciate the reviewer to this comment, which has helped us much in improving the quality of our manuscript. We have carried out some electrochemical tests to further reveal these results. The electrochemically active surface area (ESCA) of catalyst is estimated from electrochemical double-layer capacitance (C_{dl}), which is obtained by measuring double-layer charging from the cyclic voltammograms (CVs) at different scan rates in non-Faradaic potential range in 0.1 M KOH. The linear slope of charging current versus the scan rate represents C_{dl} . As shown in **Supplementary Figure 18**, the C_{dl} of Fe-SAs/NPS-HC catalyst (117.5 mF/cm²) is higher than that of Fe-SAs/NPS-C (72.3 mF/cm²), Fe-SAs/NP-C (73.0 mF/cm²), Fe-SAs/N-C (66.3 mF/cm²) and NPS-HC (64.9 mF/cm²), suggesting that Fe-SAs/NPS-HC possesses a larger ESCA. Electrochemical impedance spectroscopy (EIS) was measured in the frequency range from 10⁶ to 0.01 Hz. The Nyquist plots of EIS demonstrate that Fe-SAs/NPS-HC exhibits a much smaller semicircle diameter than the Pt/C and other reference catalysts, which represents a lower charge transfer resistance for Fe-SAs/NPS-HC, consistent with its more favorable charge transfer process. Taken together with original and complementary electrochemical tests, we think that the apparent lower Tafel slope for Fe-SAs/NPS-HC is attributed to its multiple advantages in synergistically improving ORR activity and accelerating ORR kinetics. (1) The larger electrochemically active surface area (ESCA) can expose more active sites and promote the adsorption and activation of ORR-relevant species on the Fe-SAs/NPS-HC catalyst. (2) The lower charge transfer resistance is conducive to enhancing the efficiency of charge transfer. (3) The unique structure with atomically dispersed Fe-N₄ sites and electronic control effect from the surrounding S and P atoms contribute to improving the intrinsic activity of active sites, leading to enhanced reaction efficiency of ORR-relevant species on Fe-SAs/NPS-HC, as supported by its higher mass activity and turnover frequency compared with other previously reported non-noble single-atom catalysts (**Supplementary Table 8**). (4) The hierarchical porous structure of Fe-SAs/NPS-HC improves the accessibility of active sites and facilitate the liquid water transportation and gas diffusion (*please refer to Science 2017, 357, 479; J. Power Sources 2007,*

166, 41.). Generally speaking, the unique structure functionalities and electronic control of single iron atomic sites of Fe-SAs/NPS-HC synergistically enhance ORR catalytic process, leading to the remarkable Tafel slope. Currently, it is hard to say that the apparent low Tafel slope suggests a different ORR pathway. To verify a new ORR pathway, more in-depth study needs to be carried out, such as in-situ detection of intermediate reaction species, however, it is hard for us to achieve it. We hope that our work will inspire some researchers in this field to study this important subject to promote the development of electrocatalysis.

To assess the effect of hollow structure on the ORR performance, the N, P, S co-doped solid carbon polyhedron with isolated Fe single atomic sites (Fe-SAs/NPS-C) catalyst was prepared. Compared with hollow Fe-SAs/NPS-HC, the solid Fe-SAs/NPS-C exhibits apparently degraded ORR activity and kinetics with a negatively shifted $E_{1/2}$ and smaller kinetic current density as well as larger Tafel slope (**Figure 4**). As shown in **Supplementary Figure 18**, the C_{dl} of Fe-SAs/NPS-HC (117.5 mF/cm²) is higher than that of Fe-SAs/NPS-C (72.3 mF/cm²), which suggests Fe-SAs/NPS-HC has a larger electrochemically active surface area, facilitating the adsorption and activation of ORR-relevant species on the Fe-SAs/NPS-HC. And the hollow structure with hierarchical porous of Fe-SAs/NPS-HC improves the accessibility of active sites and facilitate the liquid water transportation and gas diffusion (*please refer to Science 2017, 357, 479; J. Power Sources 2007, 166, 41.*). These results reveal that hollow structure favors mass transportation for ORR. Regarding the charge transfer, the electrochemical impedance spectroscopy (EIS) measurements were carried out. As shown in **Supplementary Figure 32**, hollow Fe-SAs/NPS-HC exhibits a lower charge transfer resistance compared with solid Fe-SAs/NPS-C, suggesting hollow structure is beneficial to charge transfer during ORR process.

➤ According to your comment, the new supplemental electrochemical tests and data have been added in the revised manuscript, as follows:

The statement “As shown in **Supplementary Figure 18**, the C_{dl} of Fe-SAs/NPS-HC catalyst (117.5 mF/cm²) is higher than that of Fe-SAs/NPS-C (72.3 mF/cm²) and NPS-HC (64.9 mF/cm²), suggesting that Fe-SAs/NPS-HC possesses a larger electrochemically active surface area (ESCA). Electrochemical impedance spectroscopy (EIS) was measured in the frequency range from 10⁶ to 0.01 Hz. The Nyquist plots of EIS demonstrate that Fe-SAs/NPS-HC exhibits a much smaller semicircle diameter than the Pt/C and other reference catalysts, which represents a lower charge transfer resistance for Fe-SAs/NPS-HC, consistent with its more favorable charge transfer process (**Fig. 4d**).” has been added in the revised manuscript (paragraph beginning in Page 11, Line 27, highlighted in **BLUE**)

The statement “The larger electrochemically active surface area can expose more active sites and promote the adsorption and activation of ORR-relevant species on the Fe-SAs/NPS-HC catalyst (**Supplementary Fig. 18**). The lower charge transfer resistance is conducive to enhancing the efficiency of charge transfer (**Supplementary Fig. 33**). The unique structure with atomically dispersed Fe-N₄ sites and electronic control effect from the surrounding S and P atoms contribute to improving the intrinsic activity of active sites, leading to enhanced reaction efficiency of ORR-relevant species on Fe-SAs/NPS-HC, as supported by its higher mass activity (MA) and turnover frequency (TOF) compared with other previously reported non-noble single-atom catalysts (**Supplementary Table 8**). The hollow structure functionalities with hierarchical porous of Fe-SAs/NPS-HC improve the accessibility of active sites and facilitate mass transport properties^{3,34}.” has been added in the revised

manuscript (paragraph beginning in Page 17, Line 22, highlighted in **BLUE**).

The corresponding **Fig. 4d**, **Supplementary Fig. 18**, **Supplementary Fig. 33** and **Supplementary Table 8** have been added in the revised manuscript and Supplementary Information.

Supplementary Figure 33 | Nyquist plots of Fe-SAs/NPS-HC and the corresponding reference catalysts in 0.1 M KOH.

Supplementary Figure 18 | Cyclic voltammograms (CVs) for ORR at different scan rates in the range of no Faradaic processes for measuring C_{dl} . (a-e) CV curves of the Fe-SAs/NPS-HC, Fe-SAs/NPS-C, Fe-SAs/NP-C, Fe-SAs/N-C and NPS-HC in 0.1 M KOH solution in the region of 0.05~0.15V vs. RHE for ORR. (f) CV fitting curves of Fe-SAs/NPS-HC, Fe-SAs/NPS-C, Fe-SAs/NP-C, Fe-SAs/N-C and NPS-HC catalysts in 0.1 M KOH for ORR.

Supplementary Table 8 | Comparison of mass activity and turnover frequency of Fe-SAs/NPS-HC with other non-noble single-atom catalysts reported in literatures.

Electrocatalysts	Kinetic				Ref.
	$E_{1/2}$ (V vs RHE)	Current Density at 0.85 V (mA/cm ²)	MA at 0.85 V (A/g)*10 ⁻³	TOF [e/(site*s)]	
Fe-SAs/NPS-HC	0.912	71.9	9.15	5.29	Our work

Fe-SAs/NPS-C	0.894	34.6	4.4	2.54	Our work
Fe-SAs/NP-C	0.881	19.7	2.51	1.45	Our work
Fe-SAs/N-C	0.87	11.7	1.49	0.861	Our work
CNT/PC (Fe)	0.88	25.7	1.1	0.638	J. Am. Chem. Soc. 2016 , 138, 15046.
S,N-Fe/N/C-CNT	0.85	7.35	0.368	0.213	Angew. Chem. Int. Ed. 2017 , 56, 610.
Fe-ISAs/CN	0.9	37.83	4.29	2.48	Angew. Chem. Int. Ed. 2017 , 56, 6937.
Fe-N4 SAs/NPC	0.885	7.47	0.747	0.432	Angew. Chem. Int. Ed. 2018 , 57,1.DOI:10.1002/anie.201804349
FeSA-N-C	0.891	23.27	4.63	2.68	Angew. Chem. Int. Ed. 2018 , 57, 1.DOI:10.1002/anie.201803262
Co SAs/N-C(900)	0.881	15.3	0.937	0.572	Angew. Chem. Int. Ed. 2016 , 55, 10800.
Co-ISAs/p-CN	0.838	2.92	1.36	0.83	Adv. Mater. 2018,30, 1706508
Cu-N-C-60	0.8	1.32	0.0507	0.0334	Energy Environ. Sci. 2016 , 9, 3736-3745.

Comment 2: The stability of single atomic Fe catalysts is impressive. Any agglomeration or leaching of the single sites? Any compositional or shape evolution of the catalyst upon cycling? Characterization of the catalysts after cycling tests is necessary.

Response: Many thanks to this comment. According to the reviewer's suggestion, a series of detailed characterizations of the used Fe-SAs/NPS-HC catalyst after cycling measurements have been performed to demonstrate the stability of the single atomic Fe catalyst as follows:

- (1) X-ray diffraction (XRD): The XRD pattern of the used Fe-SAs/NPS-HC catalyst shows there is no signals for metallic Fe species, implying no obvious agglomeration compared with the fresh catalyst. (**Supplementary Figure 21**)
- (2) Transmission electron microscopy (TEM): TEM image shows that the used Fe-SAs/NPS-HC catalyst remains the hollow dodecahedral structure. And no obvious Fe nanoparticle can be observed in the used catalyst by TEM. In addition, the corresponding mapping analysis indicates perfect retain of uniform distributions of Fe, C, N, S and P elements throughout the entire architecture of the used Fe-SAs/NPS-HC catalyst. (**Supplementary Figure 22**).
- (3) Aberration-corrected HAADF-STEM (AC HAADF-STEM): More AC HAADF-STEM images of the used catalyst show uniformly dispersed bright dots, consistent with that of fresh catalyst. AC HAADF-STEM images prove that the existence of Fe species is still in form of isolated single atoms. (**Supplementary Figure 23**)
- (4) X-ray absorption fine structure analysis (XAFS): The Fe K-edge XAENS and EXAFS spectra of the used catalyst are shown in Figure S15 and S16, indicating no obvious changes in the local structure of the single atomic Fe. (**Supplementary Figure 24**)

(5) ICP-OES analysis: The Fe loading of the used Fe-SAs/NPS-HC is 1.52 wt%, which is consistent with the fresh catalyst. Furthermore, the content of Fe in the used electrolyte after cycling tests is not detected. The above results indicate that Fe species in the used Fe-SAs/NPS-HC catalyst is not leached.

Taken together, the above results well confirm atomic dispersion of Fe species in the used Fe-SAs/NPS-HC catalyst remains unchanged. In addition, no agglomeration or leaching of Fe species in the used catalyst are detected. We have added the data in the revised manuscript as follow: “Furthermore, the detailed characterization of the used Fe-SAs/NPS-HC catalyst after durability test demonstrates the atomic dispersion of iron atoms still remains (Supplementary Fig. 21-24).” (please see: Page 12, paragraph 3)

Supplementary Figure 21 | The XRD pattern of the used Fe-SAs/NPS-HC catalyst after durability test compared with the fresh catalyst.

Supplementary Figure 22 | The structural characterizations of the used Fe-SAs/NPS-HC catalyst. (a) TEM image of the used Fe-SAs/NPS-HC. Scale bar, 500 nm. (b) The enlarged HAADF-STEM

image and corresponding EDS element maps (Fe: purple, C: blue, N: cyan, S: orange, P: green).
Scale bar, 100 nm.

Supplementary Figure 23 | The more aberration-corrected HAADF-STEM images and the enlarged images of the used Fe-SAs/NPS-HC catalyst. The single Fe atoms are marked by yellow circles (a and c: Scale bar, 5 nm; b and d: Scale bar, 1 nm).

Supplementary Figure 24 | The Fe K-edge k^2 -weighted Fourier transform (FT) spectra of the used Fe-SAs/NPS-HC (red line) and the fresh catalyst (blue line).

Comment 3: Looking at the Fe K-edge XANES curve, the profiles between Fe₂O₃ and the single atom sample are very close, particularly in the pre-peak region. This similarity implies that Fe is likely in Fe(III) valence. If so, the superior activity is out of expectation since Fe(III) species are believed to be ORR inactive compared to Fe(0). Also, fitting of the XANES spectra is quite bad. The authors are suggested to perform other analysis of the Fe state such as the useful Mossbauer spectroscopy.

Response: We appreciate the reviewer's suggestion. We have carefully checked the Fig. 3a in the manuscript. Indeed, as shown in Fig. 3a, the Fe K-edge XANES curve of Fe-SAs/NPS-HC (our single-atom sample) is close to the Fe₂O₃, indicating that the oxidation state of Fe is relatively high valence state. As shown in **Supplementary Figure 8**, the average oxidation state of Fe in the Fe-SAs/NPS-HC is near +2.5. The Fe nanoparticles, iron carbides (Fe₃C) and atomically dispersed Fe-N_x sites usually co-exist in the early reported Fe-based ORR catalysts (please refer to: *Adv. Mater.* 2012, 24, 1399-1404; *Nat. Commun.* 2013, 4, 1922; *Angew. Chem. Int. Ed.* 2013, 52, 1026-1030.). The common characterization techniques such as XRD, XPS and TEM are sensitive to Fe nanoparticles and Fe₃C, but not to atomically dispersed Fe-N_x sites, so many researchers attributed the origin of ORR activity to the existence of Fe nanoparticles and Fe₃C. With the development of advanced characterization techniques and deep research, the crucial role of atomically dispersed Fe-N_x sites on ORR performance has been discovered and verified (please refer to: *J. Am. Chem. Soc.* 2014, 136, 10882-10885; *Nat. Mater.* 2015, 14, 937-942; *Energy Environ. Sci.*, 2016, 9, 2418-2432; *Angew. Chem. Int. Ed.* 2017, 56, 6937-6941.). In our work, we prepared a single iron atomic sites (Fe-SAs/NPS-HC) catalyst to achieve high atom-utilization efficiency and full exposure of atomically dispersed Fe-N₄ active sites. Furthermore, the electron donation from the surrounding sulfur and phosphorus make the metal center Fe (Fe^{δ+}) of Fe-SAs/NPS-HC less positive, leading to enhanced ORR kinetics. The unique atomic dispersion of Fe coordinated by N and the electronic control effect from the surrounding S and P atoms of Fe-SAs/NPS-HC contribute to its superior ORR performance.

Supplementary Figure 8 | Average oxidation state of Fe in the Fe-SAs/NPS-HC. The oxidation state of Fe could be reflected through the absorption edge of Fe K-edge. The average state of the Fe-SAs/NPS-HC was ~ 2.5 , in comparison with Fe foil (0), FeO and Fe₂O₃.

(2) In order to further confirm the Fe state of our catalyst, according to the reviewer's suggestion, Mössbauer spectroscopy measurements have been carried out. Analysis of the Mössbauer spectra and hyperfine parameters of Fe-SAs/NPS-HC have been added in the revised manuscript as follows: "In order to further confirm the Fe state of Fe-SAs/NPS-HC, we performed Mössbauer spectroscopy measurements, which reveals the spectra with isomer shift of Fe³⁺ or low/intermediate spin state of Fe²⁺. Mössbauer spectra of Fe-SAs/NPS-HC at 292 K and at 78 K are presented in **Supplementary Fig. 9**, and the corresponding hyperfine parameters are listed in the **Supplementary Table 1**. The room-temperature (RT) spectra are clearly of relaxational nature revealed by slowly decaying wings (**Supplementary Fig. 9a**). From the RT-spectra fitted with a single asymmetric component, implying either magnetic Blume³¹ or electron-hopping relaxation mechanisms³², the average IS of 0.48 mm/s and 0.45 mm/s could be determined, which are clearly higher than the IS of high-spin Fe³⁺. Measurements at 78 K revealed indeed the absence of high-spin Fe³⁺ inactive in ORR (**Supplementary Fig. 9b**). Two magnetic sextet subspectra evolved at 78 K from the relaxational RT-spectra were assigned to Fe^{III}N₄ (S=1/2) and Fe^{II}N₄ (S=1) species identified by the hyperfine fields (B_{hf}) of 10.5(1) T and 17.3(1) T, respectively, according to the general rule for the saturation fields $B_{hf}^{sat}/S \approx 20$ T. Ferromagnetic ordering in the dilute system of S=1/2 and S=1 species below Curie temperature (T_C) of 215 K was observed in the temperature dependence of magnetic susceptibility (**Supplementary Fig. 10**) measured at the applied field of 1 kOe and in M-H loops at 2K, 77K and 300K (**Supplementary Fig. 11**). **Supplementary Figure 12** shows indeed that the measured B_{hf} are close to saturation B_{sat} (at 0 K). A minor (20%) doublet species was identified with low-spin Fe²⁺, because the signal from single-atom dispersed diamagnetic S=0 species may only experience the line broadening, but not Zeeman splitting below T_C. The average oxidation state of Fe in the Fe-SAs/NPS-HC is near +2.5. The axial orientations of both fields B_{hf} (parallel to V_{zz}) reveal that only Fe^{III}N₄ and Fe^{II}N₄ species are confirmed, and no other Fe-related phases (such as Fe, Fe_xC, Fe_xS and Fe_xP) are detected, indicating that the isolated Fe atoms are only coordinated by N atoms, which are consistent with the XANES analysis." (please see: page 8, paragraph 2)

Supplementary Figure 9 | Mössbauer spectroscopy of Fe-SAs/NPS-HC. (a) The room-temperature (RT) spectrum exhibits the slowly decaying wings indicating the relaxational nature of two magnetically collapsed subspectra complemented with a minor paramagnetic doublet D1; (b) The spectrum measured at 78 K consist of the same minor doublet and two magnetically split subspectra with mixed (M+Q) interaction, both showing the large quadrupole lineshifts identifying the orientation of the hyperfine fields normal to the plane of Fe-N₄, i.e. $H \parallel V_{zz}$ with positive values of V_{zz} for both Fe(II) and Fe(III) sites ($V_{zz} \perp \text{Fe(II)-N}_4$ and $V_{zz} \perp \text{Fe(III)-N}_4$).

Supplementary Figure 10 | Temperature dependence of magnetic susceptibility in the applied field of 1 kOe for the Fe-SAs/NPS-HC, showing the ferromagnetic ordering in the system of S=1 and S=1/2 species below T_C . The Curie temperature of 215 K was determined by fitting a sigmoidal function to the $\chi(T)$ curves.

Supplementary Figure 11 | M-H loops measured below (2K and 77K) and above (at 300 K) of the Curie point of 215K for the Fe-SAs/NPS-HC. The magnetic hysteresis at 77K is characterized by the saturation remanence of 0.5 emu/g and of coercive force of 5000 Oe.

Supplementary Figure 12 | Hyperfine fields in the Fe-SAs/NPS-HC at 77K and at 130 K. The dotted lines are the eye guides to show the saturation fields (at 0 K).

Supplementary Table 1 | Hyperfine parameters derived from Mössbauer spectra of Fe-SAs/NPS-HC at 292 K and at 78 K: the isomer shifts δ_{iso} , quadrupole splitting ΔE_Q , quadrupole lineshifts ϵ , internal fields B_{hf} , electric field gradient (EFG) principal components V_{zz} , angles Θ between V_{zz} and B_{hf} , linewidth LW, and subspectra areas.

	T (K)	δ_{iso} (mm/s)	B_{hf} (T)	ϵ^{**} (sites 1, 2) ΔE_Q (site 3) (mm/s)	V_{zz} (10^{21}V/m^2)	Θ ($^\circ$)	LW (mm/s)	Area (%)	Assignment
Site 1	292	0.41(5)	Relax.*	0.52(0 ^{***})	3.11	0	0.26(0)	29	$\text{Fe}^{\text{II}}\text{N}_4$, middle-spin
	78	0.48(2)	17.3(1)	0.52(4)			0.44(5)	31	
Site 2	292	0.40(1)	Relax.*	0.18(0 ^{***})	1.05	0	0.26(0)	51	$\text{Fe}^{\text{III}}\text{N}_4$, low-spin
	78	0.45(1)	10.5(1)	0.18(2)			0.46(4)	50	
Site 3	292	0.508(6)	0	0.71(4)	undefined	—	0.26(0)	20	$\text{Fe}^{\text{II}}\text{N}_4$, low-spin
	78	0.54(2)	0	0.90(4)	undefined	—	0.49(6)	19	

Notes: Θ is the polar angle of the hyperfine field B_{hf} in the reference frame of the principal EFG axes (V_{xx}, V_{yy}, V_{zz}). Because Θ was fitted to be ≈ 0 , the other polar coordinate (angle ϕ) and asymmetry parameter η are undefined, see Eq. (1) below.

*Singlet-shaped collapsed subspectra with slowly decaying wings fitted with magnetic relaxation at the rates of 8 to 8.5 Mrad/s.

** ϵ is the first-order quadrupole lineshift related to the quadrupole energy $\Delta E_Q = e^2qQ/2$ via Θ , ϕ and $\eta = (V_{xx} - V_{yy})/V_{zz}$:

$$\epsilon = \frac{\Delta E_Q}{4} (3 \cos^2 \theta - 1 + \eta \sin^2 \theta \cos 2\phi) \quad (1)$$

*** Fixed to be similar to the values obtained at 78K.

Comment 4: In DFT calculation of the OER process, why U=0 V and 1.0 V are selected? I'd suggest computation at the thermodynamic equilibrium potential of ORR, i.e., U=1.23V.

Response: We appreciate the reviewer to this comment. (1) We are sorry that we did not write clearly enough in our former manuscript. Density functional theory (DFT) calculations in our manuscript were based on the real testing conditions. That is to say, we considered the impact of pH condition (pH=13) (Reference: *J. Am. Chem. Soc.* 2014, 136, 4394-4403.; *J. Phys. Chem. B* 2004, 108, 46, 17886-17892). As mentioned in **Computational details**, reaction free energy is further calculated by the equation: $\Delta G(U, \text{pH}, p = 1 \text{ bar}, T = 298 \text{ K}) = \Delta G_0 + \Delta G_{\text{pH}} + \Delta G_U$, where ΔG_{pH} is the correction of the free energy of H^+ -ions at a pH different from 0: $\Delta G_{\text{pH}} = -kT \ln[\text{H}^+] = kT \ln 10 \times \text{pH}$. $\Delta G_U = -n\text{e}U$, where U is the applied electrode potential, n is the number of electrons transferred. Hence, the equilibrium potential U^0 for ORR at pH = 13 was determined to be 0.462 V (vs normal hydrogen electrode, NHE). As shown in Figure 5a, 0 V is without applied voltages on the basis of pH condition. And 1 V is the onset potential on the basis of pH condition. The formula of U^{RHE} as follows: $U^{\text{RHE}} = U^{\text{NHE}} + 0.0591\text{pH}$.

(2) According to the reviewer's suggestion, we have added the computation at the thermodynamic equilibrium potential ($U^{\text{RHE}} = 1.23\text{V}$) of ORR in the revised manuscript. The sentence has been added in the revised manuscript as follows: "To further elucidate the roles of electronic nature of the metal centers, the calculated free energy diagram on different doped types of Fe-SAs samples at 1.23 V (thermodynamic equilibrium potential of ORR) was presented in **Supplementary Information Figure 31**. The last two electrochemical steps of Fe-SAs/N-C and Fe-SAs/NP-C are more endothermic than Fe-SAs/NPS-C at $U^{\text{RHE}} = 1.23 \text{ V}$. The binding energy of Fe-SAs/NPS-C for the intermediates is the weakest than that of Fe-SAs/N-C and Fe-SAs/NP-C, indicating that Fe-SAs/NPS-C exhibits the highest reactivity than Fe-SAs/N-C and Fe-SAs/NP-C. The result is consistent with the results of computation at the $U^{\text{RHE}} = 0 \text{ V}$ and 1.0 V vs RHE at the pH=13 condition." (please see: starting in Page 15, the last line)

Supplementary Figure 31. Free energy diagram of ORR at $U^{\text{RHE}} = 1.23 \text{ V}$ on different doped types of Fe-SAs/NPS-C, Fe-SAs/N-C and Fe-SAs/NP-C. The last two electrochemical steps of Fe-SAs/N-C and Fe-SAs/NP-C are more endothermic than Fe-SAs/NPS-C at $U^{\text{RHE}} = 1.23 \text{ V}$, suggesting that Fe-SAs/N-C and Fe-SAs/NP-C exhibit less reactivity compared with Fe-SAs/NPS-C.

Comment 5: Applying K-L equation to determine electron transfer number in ORR sometimes gives rise to abnormal value. I would recommend including a comparison of Pt disk electrode data. RRDE tests are useful to calibrate n values as well.

Response: We appreciate the reviewer to this comment, which has helped us much in improving the quality of our manuscript. We have performed the RDE measurements to obtain the LSV curves of the commercial Pt/C at various rotation rates (Figure S19). The K-L plots for the Pt/C at different potential shows nearly parallel fitting lines. And the corresponding electron-transfer number (n) is calculated to be in the range 3.93-4.01 by applying the K-L equation. Furthermore, the RRDE tests have been carried out to assess the ORR selectivity for H₂O₂ yield of Fe-SAs/NPS-HC and the Pt/C. As shown in Figure 4f, the H₂O₂ yield of Fe-SAs/NPS-HC remains below 4.2 % in the potential range 0.3-0.9 V, corresponding to a high electron-transfer number (n) of 3.90-4.00. For comparison, the n value of 3.93-4.00 is achieved for the Pt/C. The sentence has been added in the revised manuscript as follows: “Moreover, the rotating ring disk electrode (RRDE) tests reveal the H₂O₂ yield of Fe-SAs/NPS-HC remains below 4.2 % in the potential range 0.3-0.9V (Fig. 4f), corresponding to a high electron-transfer number of 3.90-4.00. For comparison, the n value of 3.93-4.00 is achieved for the Pt/C.” (please see: Page 12, paragraph 2)

Supplementary Figure 19 | The ORR polarization curves of 20% Pt/C at different rotating rates (inset: K-L plots and electron transfer numbers).

Figure 4(f) Electron transfer number (n) (top) and H_2O_2 yield (bottom) versus potential.

Response to Reviewer #3 (Remarks to the Author):

The authors reported a N, P, S co-doped atomic Fe based carbon catalyst for efficient oxygen reduction reaction. The doped P and S are proposed to enhance the activity of Fe-N-C configuration. This article provide a new sight that the multi dopants could improve the performance of atomic metal species. The manuscript is well written and recommended to be published on Nature Communications after several issues being addressed.

Comment 1: Atomic metal based catalysts were reported to be active for HER, OER and ORR, especially for the four coordinated atomic metals. Two important publication in this field should be cited (CHEM, 2018, 4, 285; Mater. Chem. Front., 2018, 2, DOI: 10.1039/c8qm00070k).

Response: We appreciate your pointing these references to us. According to your suggestion, two important references you mentioned have been cited in the right place (ref 20 and 25) in our revised manuscript.

Comment 2: The authors should give the content of each element in the sample Fe-SAs/NPS-HC, since from the XPS spectra of the C 1s, the fractions of C-P, C-S and C-N are comparable to C=C indicating a large amount of heteroatom dopants. The content of element is an important information to assist further constructing the models for DFT calculations.

Response: Thanks for the reviewer's critical and insightful comment. As mentioned in our manuscript, the content of Fe in Fe-SAs/NPS-HC is determined to be 1.54 wt% by ICP-OES. Based on X-ray photoelectron spectroscopy (XPS) analysis, the element atomic content of Fe, N, P, S, O and C is calculated to be 0.48 at%, 6.45 at%, 1.91 at%, 1.79 at%, 3.55 at%, and 86.82 at%, respectively (Supplementary Fig. 15). And the corresponding element atomic ratio of Fe, N, P and S is calculated to be 1:13.43:3.98:3.73, which is close to the ratio of 0.5:7:2:2.

Supplementary Figure 15. The percentages of Fe, N, P, S, O and C in Fe-SAs/NPS-HC measured by XPS analysis.

We agree with the reviewer's statement that the content of element is an important information to assist further constructing the models for DFT calculations. Actually, when we constructed the models for DFT calculations, we have considered the element atomic ratio. We suppose that Fe atoms have occupied half of anchoring sites. And we have determined the atomic ratio of N, P and S element as 7:2:2, which is consistent with the XPS results. As shown in **Supplementary Figure 15**, the element atomic ratio of Fe, N, P, S is determined as 1:7:2:2 in the computational model of Fe-SAs/NPS-HC is in agreement with the above analysis. It is worth mentioning that this computational model represents the surrounding coordination environment of Fe-N₄ active sites. O element in C-based materials is generally considered to be inert for enhancing ORR performance and the effect of O element on ORR performance is ruled out in most of the previously reported literatures (please refer to: *Angew. Chem. Int. Ed.*, **2017**, 56, 13800; *J. Am. Chem. Soc.*, **2016**, 138, 15046; *PNAS*, **2018**, 115, 6626.). Therefore, O element has not been considered to be included in our computational model, as reported in the references. The sentence has been added in the revised manuscript as follows: "Based on the hyperfine parameters of Mössbauer spectra and the best-fitting EXAFS results as well as the element atomic content analysis by XPS (**Supplementary Fig. 15**), DFT calculations construct and optimize the structural model of Fe-SAs/NPS-HC (Fig. 3e), which is consistent with the findings of EXAFS and XANES measurements." (please see: Page 9, paragraph 2)

Comment 3: From the XPS spectrums of P 2p and N 1s, oxygen is confirmed in the sample. Does oxygen have any effects to the final ORR performance (being involved or accelerating the reaction)?

Response: Thanks for this suggestion. Based on the XPS analysis, the presence of O element is confirmed. The O element in C-based materials is generally considered to be inert for enhancing ORR performance. In the most of the reported references, O element is excluded in the identification of active species of ORR (please refer to: *Angew. Chem. Int. Ed.*, **2017**, 56, 13800; *J. Am. Chem. Soc.*, **2016**, 138, 15046; *PNAS*, **2018**, 115, 6626.). In addition, when researchers construct the DFT models to investigate catalytic mechanism of ORR, they usually ignore the presence of O element. As we can see in most reported references, the O element is not contained in the DFT models, even though a large amount of O species exists in C-based materials. Therefore, we have not discussed the effect of O element on ORR performance.

On the other hand, whether or not O element has some effects on ORR performance, its existence in Fe-SAs/NPS-HC catalyst will hardly affect the key statements in our work that "Functionalized hollow structure plays a vital role in enhancing ORR kinetics." and "The unique atomic dispersion of Fe coordinated by N atoms and the electronic control effect from the surrounding S and P atoms contribute to high efficiency and satisfied kinetics for ORR." For example, as shown in Table R1, based on the XPS analysis, the element atomic content of O in solid N, P, S co-doped single-atom Fe (Fe-SAs/NPS-C) catalyst is almost similar to that in hollow N, P, S co-doped single-atom Fe (Fe-SAs/NPS-HC) catalyst. Therefore, whether or not O element has some effects on ORR performance has little influence on the investigation of the impact of hollow structure on ORR performance.

Table R1. The element atomic content in Fe-SAs/NPS-HC and Fe-SAs/NPS-C.

Sample	Element atomic content (%)					
	Fe	N	P	S	O	C
Fe-SAs/NSP-HC	0.48	6.45	1.91	1.79	3.55	86.82
Fe-SAs/NSP-C	0.44	6.84	1.74	1.93	3.17	85.88

Comment 4: The authors verified the positive effect to the ORR performance of the hollow structure by comparing the performances of Fe-SAs/NPS-HC and the corresponding bulk sample. However, is this bulk sample has the similar component (amount of each element) with Fe-SAs/NPS-HC?

Response: Thanks for this comment. Actually, to make this comparison more reasonable, we have considered to design the solid N, P, S co-doped single-atom Fe (Fe-SAs/NPS-C) catalyst with almost similar component with Fe-SAs/NPS-HC catalyst. As revealed by ICP-OES, the content of Fe in Fe-SAs/NPS-C is determined to be 1.41 wt%, while the content of Fe in Fe-SAs/NPS-HC is determined to be 1.54 wt%. Given that the Fe species is the main active species for ORR, as described in the “Electrochemical measurement” section, a certain amount of samples was prepared and dispersed in in 1 mL of a mixture solution containing ethanol (0.495 mL), water (0.495 mL) and 0.5 wt% Nafion solution (10 μ L) to make the corresponding catalyst inks with the same Fe content as Fe-SAs/NPS-HC. Therefore, the content of Fe in the test inks of Fe-SAs/NPS-C is the same as that of Fe-SAs/NPS-HC. As shown in Table R1, the element atomic content of Fe, N, P, S, O and C in solid Fe-SAs/NPS-C catalyst is almost similar with that of hollow Fe-SAs/NPS-HC catalyst.

Table R1. The element atomic content in Fe-SAs/NPS-HC and Fe-SAs/NPS-C.

Sample	Element atomic content (%)					
	Fe	N	P	S	O	C
Fe-SAs/NSP-HC	0.48	6.45	1.91	1.79	3.55	86.82
Fe-SAs/NSP-C	0.44	6.84	1.74	1.93	3.17	85.88

Comment 5: Please calculate the mass activity and turnover frequency of Fe-SAs/NPS-HC and compare with other atomic metal based catalysts. Normally, atomic metal based catalysts possess large mass activities and turnover frequencies.

Response: We thank the reviewer for this comment. The mass activity and turnover frequency of Fe-SAs/NPS-HC have been calculated and compared with other non-noble single-atom catalysts reported in literatures (**Supplementary Table 8**). The mass activities (MAs) and turnover frequencies (TOFs) of the catalysts were calculated by normalizing them with total metal content according to the following equations:

$$\text{MA [A/g]} = \frac{j_k}{m_0 \times w_{\text{metal}}}$$

$$\text{TOF [e/(site}\cdot\text{s)]} = \frac{j_k}{n_{\text{metal}} \times F} = \frac{j_k \times M_{\text{metal}}}{m_{\text{metal}} \times F} = \frac{j_k \times M_{\text{metal}}}{m_0 \times w_{\text{metal}} \times F}$$

Where j_k is the kinetic current, m_0 is the mass loading of catalysts on glassy carbon electrode surface, w_{metal} is the mass concentration of metal in the catalysts, n_{metal} is the mol amount of metal in the catalysts, m_{metal} is the mass of metal in the catalysts, M_{metal} is the atomic weight of metal,

and F is the Faraday constant (96485 C/mol).

The above-mentioned equations and the corresponding Table have been added in the revised Supplementary Information. And the statement “The unique structure with atomically dispersed Fe-N₄ sites and electronic control effect from the surrounding S and P atoms contribute to improving the intrinsic activity of active sites, leading to enhanced reaction efficiency of ORR-relevant species on Fe-SAs/NPS-HC, as supported by its higher mass activity (MA) and turnover frequency (TOF) compared with other previously reported non-noble single-atom catalysts (Supplementary Table 8).” has been added in the revised manuscript.(please see: Page 16, paragraph 2)

Supplementary Table 8. Comparison of mass activity and turnover frequency of Fe-SAs/NPS-HC with other non-noble single-atom catalysts reported in literatures.

Electrocatalysts	$E_{1/2}$ (V vs RHE)	Kinetic Current Density at 0.85 V (mA/cm ²)	MA at 0.85 V (A/g)*10 ⁻³	TOF [e/(site*s)]	Ref.
Fe-SAs/NPS-HC	0.912	71.9	9.15	5.29	Our work
Fe-SAs/NPS-C	0.894	34.6	4.4	2.54	Our work
Fe-SAs/NP-C	0.881	19.7	2.51	1.45	Our work
Fe-SAs/N-C	0.87	11.7	1.49	0.861	Our work
CNT/PC (Fe)	0.88	25.7	1.1	0.638	J. Am. Chem. Soc. 2016, 138, 15046.
S,N-Fe/N/C-CNT	0.85	7.35	0.368	0.213	Angew. Chem. Int. Ed. 2017, 56, 610.
Fe-ISAs/CN	0.9	37.83	4.29	2.48	Angew. Chem. Int. Ed. 2017, 56, 6937.
Fe-N ₄ SAs/NPC	0.885	7.47	0.747	0.432	Angew. Chem. Int. Ed. 2018, 57, 1.DOI:10.1002/anie.201804349
FeSA-N-C	0.891	23.27	4.63	2.68	Angew. Chem. Int. Ed. 2018, 57, 1.DOI:10.1002/anie.201803262
Co SAs/N-C(900)	0.881	15.3	0.937	0.572	Angew. Chem. Int. Ed. 2016, 55, 10800.
Co-ISAS/p-CN	0.838	2.92	1.36	0.83	Adv. Mater. 2018,30, 1706508

Cu-N-C-60	0.8	1.32	0.0507	0.0334	Energy Environ. Sci. 2016, 9, 3736-3745.
-----------	-----	------	--------	--------	---

Comment 6: The authors constructed a DFT model of Fe-SAs/NPS-HC. How to identify this model in the sample? Is there any direct or indirect evidence to guide setting up this model?

Response: Thanks for the reviewer's critical and insightful comment. We have considered this point very carefully. At first, we construct the DFT model of Fe-SAs/NPS-HC catalyst by synthetically considering the aspects as follows:

- (1) In terms of atomic structure characterizations, we have performed aberration-corrected HAADF-STEM (AC HAADF-STEM), X-ray absorption fine structure (XAFS) and the wavelet transform (WT) measurements to identify the structure of Fe-SAs/NPS-HC. As given in AC HAADF-STEM analysis in the manuscript (**Figure 1e,f**), the corresponding bright dots have been observed, consistent with the isolated single Fe atoms. Only one peak at the distance of ~ 1.5 Å in Fe K-edge EXAFS Fourier transform curve of Fe-SAs/NPS-HC is detected, ascribed to Fe-N coordination. As shown in Figure 3c, only one intensity maximum at approximately 4.2 Å⁻¹ is clearly exhibited, suggesting the contribution of Fe-N coordination. Further, as evidenced by quantitative least-squares EXAFS curve-fitting analysis, the EXAFS fitting curve and the corresponding fitting parameters are summarized in Figure 3d Supplementary Table 1. The best-fitting results clearly demonstrate that the first shell peak at 1.5 Å is ascribed to the isolated Fe site coordinated four N atoms as Fe-N₄ structure in comparison with the fitting results for Fe foil and Fe₂O₃. In addition, an average Fe-N coordination number of 3.9 and the corresponding mean bond length of ~ 2.0 Å are obtained by XAFS analysis.
- (2) In terms of the analysis of element atomic content, X-ray photoelectron spectroscopy (XPS) measurements are carried out. The element atomic content of Fe, N, P, S, O and C is calculated to be 0.48 at%, 6.45 at%, 1.91 at%, 1.79 at%, 3.55 at%, and 86.82 at%, respectively (**Supplementary Figure 15**). And the corresponding element atomic ratio of Fe, N, P and S is calculated to be 1:13.43:3.98:3.73, which is close to the ratio of 0.5:7:2:2. We suppose that Fe atoms have occupied half of anchoring sites. Therefore, we have determined the atomic ratio of Fe, N, P and S element as 1:7:2:2 in the computational model of Fe-SAs/NPS-HC.
- (3) In addition, in terms of the Fe state of the Fe-SAs/NPS-HC catalyst, Mössbauer spectroscopy measurements have been carried out. Analysis of the Mössbauer spectra and hyperfine parameters of Fe-SAs/NPS-HC have been added in the revised manuscript (*please see*: Page 8, paragraph 2). And the corresponding Supplementary Fig.9-12 and Supplementary Table 1 have been added in Supplementary Information. The Mössbauer spectroscopy reveals that only Fe^{III}N₄ and Fe^{II}N₄ species are confirmed, however, no other Fe-related phases (such as Fe, Fe_xC, Fe_xS and Fe_xP) are detected, which are consistent with the XAFS analysis. These results are strong proofs to confirm reasonability of the computational model of Fe-SAs/NPS-HC.
- (4) In summary, when we constructed the models for DFT calculations, we have considered atomic structure of Fe species and the element atomic ratio in the Fe-SAs/NPS-HC catalyst.

According to the reviewer's comment, we have considered this point very carefully and carried out the Mössbauer measurements to further confirm the Fe state of the Fe-SAs/NPS-HC catalyst. All the results give strong evidences to guide setting up the computational model of Fe-SAs/NPS-HC and confirm reasonability of this model.

Comment 7: The Fe-N coordination structure is deduced from the XAS characterization. However, do Fe-S and Fe-P coordination structures also exist in the sample?

Response: We thank the reviewer for this comment. (1) Based on this comment, we carefully examined the XAFS analysis and investigated many relevant literatures. From the Fourier transform (FT) curves, the distance of Fe-N shell is located at about 1.5 Å. The distance of the Fe-S coordination is at ~1.8 Å, which is higher than that of Fe-N coordination (~1.5 Å) (please refer to: *Angew. Chem. Int. Ed.* 2017, 56, 610-614.). Because the atomic radius of P atom is larger than that of S atom, the distance of Fe-P shell is longer than that of Fe-S shell. As shown in Fig. 3c, only one peak at the distance of ~1.5 Å in Fe K-edge EXAFS Fourier transform curve of Fe-SAs/NPS-HC is detected, ascribed to Fe-N coordination. And other higher shell peak at the distance of ~2 Å is not observed. The result confirms that there are no Fe-S and Fe-P contribution in the Fe-SAs/NPS-HC catalyst.

(2) In addition, according to analysis of the Mössbauer spectra and hyperfine parameters of Fe-SAs/NPS-HC in the revised manuscript (*please see*: page 8, paragraph 2 and corresponding discussion in Page 9 of Supplementary Information and Supplementary Table 1). The Mössbauer spectrum of Fe-SAs/NPS-HC was fitted with two magnetic sextet subspectra evolved at 78 K from the relaxational RT-spectra, which were assigned to Fe^{III}N₄ (S=1/2) and Fe^{II}N₄ (S=1) species, indicating that the Fe atoms are only coordinated by N atoms (please refer to: *Angew. Chem. Int. Ed.* 2017, 56, 13800–13804). However, no other Fe-related phases (e.g. Fe, Fe_xC, Fe_xS, Fe_xP) were observed. The findings are consistent with the XAFS analysis. All results are strong proofs to confirm that no Fe-S and Fe-P coordination structures in the Fe-SAs/NPS-HC catalyst.

Comment 8: Since the catalyst Fe-SAs/NPS-HC has a remarkable ORR activity in the alkaline media, did the authors try it in the acidic media?

Response: We thank the reviewer for this comment. We have evaluated the ORR performance of Fe-SAs/NPS-HC and reference catalysts in the acidic media by measuring in 0.5 M H₂SO₄ solution. The statement “The Fe-SAs/NPS-HC also exhibits outstanding ORR performance in acidic media (**Supplementary Fig. 25**). By RDE measurements in 0.5 M H₂SO₄ solution, an excellent ORR activity with a $E_{1/2}$ of 0.791 V is observed in Fe-SAs/NPS-HC, which approaches that of the commercial Pt/C (0.800 V), and is much better than those of Fe-SAs/NPS-C and NPS-HC (**Supplementary Fig. 25a and b**). The favorable ORR kinetics of Fe-SAs/NPS-HC is verified by higher kinetic current density of 18.8 mA cm⁻² and lower Tafel slope of 54 mV dec⁻¹ compared to Pt/C and Fe-SAs/NPS-C and NPS-HC (**Supplementary Fig. 25c**). As shown in **Supplementary Table 4**, the ORR performance of Fe-SAs/NPS-HC surpasses that of most reported non-precious metal ORR electrocatalysts. RRDE measurements demonstrate that Fe-SAs/NPS-HC exhibits a low H₂O₂ yield below 2.8% with a high electron transfer number of 3.95, suggesting its high-efficiency 4e⁻ ORR pathway (**Supplementary Fig. 25d and 26**). The accelerated durability test reveals Fe-SAs/NPS-HC shows an outstanding stability with a little

negative $E_{1/2}$ shift of 5 mV after 5000 cycles (Supplementary Fig. 25e). Moreover, Fe-SAs/NPS-HC has a superior tolerance to carbon monoxide and methanol crossover compared to Pt/C (Supplementary Fig. 25f and Supplementary Figure 27).” has been added in the revised manuscript. (please see: Page 12, paragraph 4)

Supplementary Figure 25 | Electrochemical ORR performance of Fe-SAs/NPS-HC in 0.5 M H₂SO₄ solution. (a) ORR polarization curves for Fe-SAs/NPS-HC, Fe-SAs/NPS-C, NPS-HC and 20% Pt/C. (b) Comparison of J_k at 0.75 V and $E_{1/2}$ of Fe-SAs/NPS-HC and the corresponding reference catalysts. (c) The Tafel plots for Fe-SAs/NPS-HC and the corresponding reference catalysts. (d) Electron transfer number (n) (top) and H₂O₂ yield (bottom) vs potential. (e) ORR polarization curves of Fe-SAs/NPS-HC before and after 5000 potential cycles. (f) Tolerance to CO poison of Fe-SAs/NPS-HC compared with 20% Pt/C at 0.48 V (vs. RHE). CO (flow 50 mL·s⁻¹) is injected into the 0.5 M H₂SO₄ solution at the time of 300 s and stopped at the time of 600 s.

Supplementary Figure 26 | Electrochemical ORR performance of Fe-SAs/NPS-HC in 0.5 M H₂SO₄ solution. (a) The ORR polarization curves at different rotating rates of Fe-SAs/NPS-HC **(b)** K-L plots and electron transfer numbers.

Supplementary Figure 27 | Tolerance to methanol of Fe-SAs/NPS-HC compared with the 20% Pt/C at 0.45 V (vs. RHE) in 0.5 M H₂SO₄ solution.

With these changes, we sincerely hope that you find our manuscript suitable for publication in the *Nature Communications*.

Thank you very much for your consideration.

Sincerely yours,

With best regards,
Prof. Dingsheng Wang

Reviewers' Comments:

Reviewer #2:

Remarks to the Author:

After carefully reading this updated manuscript and the authors' reply, I admire the authors sincere revision by supplementing a set of new data, but I regret to suggest that the manuscript cannot be accepted for publication in its present form. First of all, I now agree with the concern of Reviewer #1 on the scientific novelty. Pyrolysis of MOFs and polymers to synthesize heteroatom M-N-C as well as single atom catalyst are well known. The authors should state clearly what is new and what is different in this synthesis as compared with reported strategies, rather than putting together several intensively investigated methodologies to form an unreported material. Although the titled material is noble-metal free, the preparation is complicated and involves synthesis of MOF precursor, polymerization, and heat treatment. The tedious procedures and gas release make the synthetic method not economical or environmental benign. Furthermore, there are several critical scientific questions to be addressed.

1, Can the atomic ratio of Fe:N:S:P:C be controllable? Which element is the key component? How about the effect of the composition on the electrocatalytic performance?

2, Note that the pyrolysis proceeds at 900degree. In such a high temperature, why Fe, S, and P remains but Zn vaporize? The authors claimed that Fe is coordinated by N while there are no Fe-S and Fe-P contribution. Then, what are the chemical states of S and P? How can S and P withstand the high temperature pyrolysis even in the case of Zn vaporization?

Overall, I would suggest a further thorough revision to improve the quality of the manuscript.

Reviewer #3:

Remarks to the Author:

The authors answers all of my questions with satisfactory. I have no further questions and would like to recommend its publication in Adv Mater in the current version.

Response to Referees' Comments

We thank the referees for their valuable comments and positive endorsement to our manuscript.

We have carefully considered the referees' comments and revised the manuscript accordingly. Our responses and corresponding revisions are as follows:

Response to Reviewer #2 (Remarks to the Author):

General Comments: After carefully reading this updated manuscript and the authors' reply, I admire the authors sincere revision by supplementing a set of new data, but I regret to suggest that the manuscript cannot be accepted for publication in its present form. First of all, I now agree with the concern of Reviewer #1 on the scientific novelty. Pyrolysis of MOFs and polymers to synthesize heteroatom M-N-C as well as single atom catalyst are well known. The authors should state clearly what is new and what is different in this synthesis as compared with reported strategies, rather than putting together several intensively investigated methodologies to form an unreported material. Although the titled material is noble-metal free, the preparation is complicated and involves synthesis of MOF precursor, polymerization, and heat treatment. The tedious procedures and gas release make the synthetic method not economical or environmental benign. Furthermore, there are several critical scientific questions to be addressed.

1, Can the atomic ratio of Fe:N:S:P:C be controllable? Which element is the key component? How about the effect of the composition on the electrocatalytic performance?

2, Note that the pyrolysis proceeds at 900 degree. In such a high temperature, why Fe, S, and P remains but Zn vaporize? The authors claimed that Fe is coordinated by N while there are no Fe-S and Fe-P contribution. Then, what are the chemical states of S and P? How can S and P withstand the high temperature pyrolysis even in the case of Zn vaporization?

Overall, I would suggest a further thorough revision to improve the quality of the manuscript.

Response: We thank the reviewer and appreciate the suggestion, which has helped us much in improving the quality of our manuscript. In order to more clearly demonstrate our response to the comments, we have carried out point-to-point responses, as follows:

Point-to-point Responses to the Comments:

- **Comments: Pyrolysis of MOFs and polymers to synthesize heteroatom M-N-C as well as single atom catalyst are well known. The authors should state clearly what is new and what is different in this synthesis as compared with reported strategies, rather than putting together several intensively investigated methodologies to form an unreported material.**

Response: We appreciate the reviewer's suggestion. As we know, the performance of catalysts relies on rational design and optimization of their structural and electronic properties. For ORR reaction, the catalytic efficiency and kinetics are correlated with multiple steps including the adsorption and activation of substrates, charge transfer, and the desorption of products (please

refer to: *Chem. Soc. Rev.* **2015**, 44, 2060; *Science* **2017**, 355, eaad4998). **Optimizing catalyst from only one aspect is insufficient to accomplish remarkable ORR enhancement.** Therefore, developing an effective synthetic strategy to preferentially generate uniform and atomically dispersed active sites, and simultaneously achieve electronic modification and structure functionalities is highly desirable.

Our synthetic strategy is distinguished from other methods in terms of novel ultrathin hollow structure for improved mass transport and charge transfer, and tunable heteroatoms doping for electronic modification of active centers. In order to remarkably improve ORR activity and accelerate ORR kinetics, we developed the new concepts of material design via structure functionalities and electronic control of active sites with the help of MOF@polymer synthetic strategy to **simultaneously achieve the synthesis of unique atomically dispersed active sites, the construction of functionalized hollow structure via Kirkendall effect and electronic modulation of active centers by near-range coordination with nitrogen and long-range interaction with sulfur and phosphorus.** In detail, the unique structure design of our Fe-SAs/NPS-HC contributes to enhanced ORR activity and kinetics from as-follow aspects. a) The functionalized hollow structure endow Fe-SAs/NPS-HC with large electrochemically active surface area, which can expose more active sites and promote the adsorption and activation of ORR-relevant species. b) The low charge transfer resistance enhances the efficiency of charge transfer. c) The unique structure with atomically dispersed Fe-N₄ sites and electronic control effect from the surrounding S and P atoms contribute to improving the intrinsic activity of active sites, leading to enhanced reaction efficiency of ORR-relevant species on Fe-SAs/NPS-HC. d) The hierarchical porous structure of Fe-SAs/NPS-HC improves the accessibility of active sites and facilitate the liquid water transportation and gas diffusion. The above-mentioned critical factors for ORR have not simultaneously realized yet in any MOFs-derived or polymers-derived materials.

Comparatively, conventional methods are confronted with many disadvantages that prevent them from further improving the ORR performance to achieve the substitution of the expensive Pt/C catalyst. One reason is that active sites are not primarily and preferentially formed during synthesis process (please refer to: *J. Am. Chem. Soc.* **2016**, 138, 635; *J. Am. Chem. Soc.* **2016**, 138, 15046). Most synthetic routes to M-N-C catalysts necessitate to undergo a high-temperature pyrolysis, which often lead to the coexistence of active species and a large amount of less-active metal particles or carbide (please refer to: *Nat. Commun.* **2013**, 4, 1922). Such heterogeneity in structure and composition not only contributes to unsatisfied performance due to low atom-utilization efficiency and the significant decline in the number of active sites, but also hinders the in-depth understanding of active sites and further establishment of definitive correlation with catalytic properties, which is an essential basis to guide the subsequent design and optimization of catalysts for enhanced catalytic performance.

Although modifying the electronic structure of active centers is a powerful approach to enhance catalytic properties (please refer to: *Nat. Mater.* **2016**, 15, 564; *Nat. Chem.* **2018**, 10, 974), most pyrolysis routes to M-N-C catalysts are hard to achieve electronic modulation through introducing heteroatoms modifier because of poor control over dispersion property and uniformity of heteroatoms doping. In addition, many catalysts synthesized by reported methods do not possess the hierarchically porous structure, which weakens the reaction kinetics due to poor accessibility of active sites and low mass transport as well as bad charge transfer. Although some

strategies such as hard template followed by HF etching approach have been developed to construct hollow structure (please refer to: *Nat. Commun.* **2014**, 5, 4973), it is difficult to control their structure and composition, and HF solution is dangerous, which is unfavorable for large-scale application.

Benefiting from rational protocol via structure functionalities and electronic control of active sites, single iron atomic sites (Fe-SAs/NPS-HC) catalyst exhibits satisfied ORR kinetics and excellent performance in alkaline media, which makes both Reviewer# 2 and Reviewer# 3 exhilarated, in the first round of comments, “*Chen et al. report in this manuscript a MOF@polymer strategy to synthesize single Fe atomic sites catalysts for the ORR. It’s an interesting piece of work that integrates vogue concepts such as single atom catalysis, hollow nanostructure, electronic modulation, and DFT calculation. Surprisingly, the electrocatalytic performance of the titled material is striking. A half-wave potential of 0.912 V and a Tafel slope of 36 mV/dec are unprecedented, so far as I know. Besides, the materials are clearly characterized and the manuscript is well organized.*”

Moreover, after four months of revision, we have obtained more useful and exciting findings. These new findings not only demonstrate that our catalyst exhibits great potential in the substitution of the expensive Pt-based electrocatalysts to drive the cathodic ORR in PEMFC devices and Zn-air batteries, but also provide an in-depth understanding in ORR active sites at atomic scale, which **reinforce the scientific novelty and significance of our novel synthetic strategy**, as follows:

(1) Our catalyst exhibits a great potential application in PEMFCs devices. Today, the acidic polymer electrolyte membrane fuel cell (PEMFC) is still the mainstream fuel cell technology and more close to large-scale industrialization. According to the latest fuel cell roadmap from the US Department of Energy (DOE), the “durability and cost are the primary challenges to fuel cell commercialization” (<https://www.hydrogen.energy.gov/>) (Fig. R1a). From 2000 to 2012, the cost of proton exchange membrane fuel cell (PEMFC) system dropped from US\$248 kW⁻¹ to US\$45kW⁻¹ due to the reduced platinum loading and increased power density (Fig. R1b). However, from 2012 to 2016, the price adversely increases to US\$52kW⁻¹. The reason is not only the mass activity of Pt-based catalysts have reached the bottleneck in the mass-produced automotive industry, but also the price of platinum unfavourably rose from US\$15 g⁻¹ to US\$48 g⁻¹ (please refer to: *Nat. Nano.* 2016, 11, 1020). There's no making without breaking. That is, replacing the Pt-based electrodes with non-precious metal catalysts, which retain comparable power density and durability, is most likely the fundamental approach to solve the cost obstacle and achieve the large-scale commercial application of fuel cell.

In this work, the H₂/air PEMFC performance was evaluated by using Fe-SAs/NPS-HC as a cathode catalyst layers in MEA under realistic fuel cell operations with the cathode operated on air. As shown in Fig. R2a, Fe-SAs/NPS-HC based MEA reaches a high power density of **400 mW cm⁻² at 0.40 V, reaching 98.8% of the commercial Pt/C based MEA (405 mW cm⁻²)** under identical test conditions. The fuel cells performance of Fe-SAs/NPS-HC based MEA is compared favorably to that of most reported Pt-free catalysts (Table R1). Moreover, unlike the commercial Pt/C catalyst with obvious decrease of catalytic activity, Fe-SAs/NPS-HC shows a superior **tolerance to methanol crossover and carbon monoxide** (Fig. R2b and R2c), which is beneficial to its commercial application in fuel cell devices. These results demonstrate that Fe-SAs/NPS-HC

exhibits great potential in the substitution of the expensive Pt-based electrocatalysts to drive the cathodic ORR in PEMFC devices.

Figure R1. (a) Fuel cell targets by DOE versus status. (b) Annual progression of cost estimates.

Figure R2. (a) H₂/air fuel cell polarization curves and power density plots of MEAs using Fe-SAs/NPS-HC (loading of 0.8 mg cm⁻²) and 20 wt% Pt/C (loading of 0.8 mg cm⁻²) as cathode catalysts, respectively. Membrane Nafion 211, cell 80 °C, electrode area 5 cm². (b) Tolerance to CO poison of Fe-SAs/NPS-HC compared with 20% Pt/C at 0.45 V (vs. RHE). CO (flow 50 mL·s⁻¹) is injected into the 0.5 M H₂SO₄ solution at the time of 300 s and stopped at the time of 600 s. (c) Tolerance to methanol of Fe-SAs/NPS-HC compared with the 20% Pt/C at 0.45 V (vs. RHE) in 0.5 M H₂SO₄ solution.

Table R1 | Comparison of H₂/air fuel cell performance by using Fe-SAs/NPS-HC and other non-noble metal catalysts reported in the literatures as the cathode

Catalysts	Catalyst loading (mg/cm ²)	Back pressure (MPa)	Operation temperature (°C)	Peak power density (mW/cm ²)	Ref.
Fe-SAs/NPS-HC	0.8	0.2	80	400	Our work

(CM+PANI)-Fe-C	4	0.1	80	420	Science 2017 , 357, 479-484.
Fe-N-C-Phen-PANI	4	0.138	80	380	Adv. Mater. , 2017 , 29, 1604456
Fe-P-C_Ar-NH900	4	0.138	80	335	Nano Energy 2016 , 26, 267-275.
Fe-MBZ	3.0 ± 0.5	0.4	80	330	J. Power Sources 2016 , 326, 43-49.
FePhen@MOF-ArNH ₃	3	0.25	80	380	Nat. Commun. 2015 , 6, 7343.
Fe/oPD-Mela	3	0.20	80	270	Appl. Catal., B 2014 , 158-159, 60-69.
Fe/PI-1000-III-NH ₃	4	0.2	80	320	J. Mater. Chem. A 2014 , 2, 11561-11564.
Fe/TPTZ/ZIF-8	1.14	0.1	80	300	Angew. Chem. Int. Ed. 2013 , 125, 7005-7008
Co-PPY-C	0.06 (Co loading)	0.2	80	70	Nature 2006 , 443, 63-66.

(2) **Our catalyst exhibits a great potential application in Zn-air battery.** To extend the potential application of Fe-SAs/NPS-HC in energy storage and conversion devices, the Zn-air battery was assembled by applying Fe-SAs/NPS-HC as the air cathode and zinc foil as the anode with 6 M KOH electrolyte. For comparison, the Zn-air battery by using 20 wt% Pt/C as the air cathode was also made and tested in an identical condition. As shown in Figure 6a, the Fe-SAs/NPS-HC based battery exhibits a higher open circuit voltage of 1.45 V compared with the Pt/C based battery, suggesting the Fe-SAs/NPS-HC based battery can output higher voltage. The maximum power density of Fe-SAs/NPS-HC based battery achieves as high as 195.0 mW cm⁻² with a high current density of 375 mA cm⁻² (Figure 6b). Both power density and current density of Fe-SAs/NPS-HC based battery outperforms that of Pt/C based batter (177.7 mW cm⁻², 283 mA cm⁻²). The rechargeability and cyclic durability of catalysts are of great significance for the practical applications of Zn-air battery. As shown in Figure 6c, Fe-SAs/NPS-HC based battery exhibits an initial charge potential of 2.07 V and a discharge potential of 1.11 V. Obviously, Fe-SAs/NPS-HC based battery delivers a lower charge-discharge voltage gap compared with the Pt/C based battery, indicating the better rechargeability of Fe-SAs/NPS-HC based battery. Unlike the Pt/C based battery with obvious voltage change, Fe-SAs/NPS-HC based battery exhibits the negligible variation in voltage after 500 charging/discharging cycle tests with 200000 s,

suggesting the outstanding long-term durability.

(3) We have significantly advanced the scientific understanding on the ORR active sites at atomic scale through *in situ* monitoring of electronic structure of active sites under ORR operating conditions by *in situ* XAFS measurement. *In situ* XAFS is carried out to investigate the electronic structure of active sites under ORR operating conditions (Fig. R3). Fe K-edge XANES of *in situ* electrode above the onset potential such as at 1.0 V is similar to that of *ex situ* electrode, suggesting the excellent stability of active sites in electrolyte. With the applied potential from 1.0 V to 0.3 V to increase the ORR current density, the Fe K-edge XANES shifts toward lower energy, which indicates that the oxidation state of Fe species gradually decreases. The *in situ* XAFS analysis clearly reveals that the ORR performance closely correlates with the electronic structure of active sites. This finding is consistent with our designed experimental results and our DFT calculations, which demonstrates that the electron donation from the surrounding sulfur and phosphorus can make the charge of metal center Fe ($\text{Fe}^{\delta+}$) of Fe-SAs/NPS-C become less positive, which contributes to a weakened binding of the adsorbed OH species, resulting in the enhanced ORR kinetics and efficiency.

Figure R3. (a) Fe K-edge XANES spectra of Fe-SAs/NPS-HC at various potentials during ORR catalysis in O_2 -saturated 0.1 M KOH (inset: the enlarged Fe K-edge XANES spectra). (b) Proposed reaction scheme with the intermediates having optimized geometry of Fe-SAs/NPS-HC toward ORR.

- **Comment:** Although the titled material is noble-metal free, the preparation is complicated and involves synthesis of MOF precursor, polymerization, and heat treatment. The tedious procedures and gas release make the synthetic method not economical or environmental benign.

Response: In our synthetic method, the synthesis of MOF precursor and polymerization are carried out under room temperature with stirring, which is easy to operate and suitable for scale-up preparation. Therefore, our synthetic method is easily scaled up to grams of Fe-SAs/NPS-HC catalyst through a simple pyrolysis process, which is only limited by the size of the tube furnace. The large-scale preparation of Fe-SAs/NPS-HC is demonstrated in Supplementary Fig. 43. The feature of mass production of Fe-SAs/NPS-HC catalyst is highly desirable for its further commercial application in energy conversion and storage devices as the alternative of the expensive Pt-based catalysts. Moreover, we extended our synthetic method to other metal

single-atom catalysts. We have successfully synthesized various M-SAs/NPS-HC catalysts (M=Ni, Cu, Ru, Ir) through similar synthetic procedures, just by employing other metal precursors ($\text{Ni}(\text{NO}_3)_2$, $\text{Cu}(\text{NO}_3)_2$, RuCl_3 and IrCl_4) to replace $\text{Fe}(\text{NO}_3)_3$. The TEM images (Supplementary Figs. 44a, 45a, 46a and 47a) demonstrate that these M-SAs/NPS-HC catalysts displays hollow polyhedral morphologies. Only one broad peak at about 25° indexed to the graphitic carbon is observed, and no reflections ascribed to metallic nanoparticles are detected (Supplementary Figs. 44b, 45b, 46b and 47b). HAADF-STEM images and the corresponding EDS mappings of M-SAs/NPS-HC catalysts reveal the uniform dispersion of metal elements throughout the entire hollow shells (Supplementary Figs. 44c, 45c, 46c and 47c). The atomic dispersion of metal atoms in these M-SAs/NPS-HC catalysts is confirmed by AC HAADF-STEM images (Supplementary Figs. 44d, 45d, 46d and 47d), where several bright dots associated with isolated metal atoms can be clearly observed. These results demonstrate **the generality of our synthetic method**. It is worth mentioning that **our pyrolysis equipment is safe and professional. And we have employed tail gas treatment device in the pyrolysis process to exterminate or reduce environment pollution** (Fig. R4).

- We appreciate the reviewer's comments and suggestions, which help us to improve the quality and theme of our work and inspire us to think about the advantages of our synthetic methods and the highlights of our work. Therefore, we have carefully revised our manuscript (paragraph beginning in Page 2, Line 8, highlighted in **RED**).
- The large-scale preparation of catalysts is highly desirable for further commercial application. To demonstrate the mass production, grams of Fe-SAs/NPS-HC have been prepared, indicating our synthetic method is easily scaled up. This part has been added in revised manuscript (Page 19, highlighted in **RED**). The corresponding Supplementary Fig. 43 have been added in revised Supplementary Information.
- To demonstrate the generality of our synthetic method, we have extended our synthetic method to other metal single-atom catalysts, and we have achieved the synthesis of various M-SAs/NPS-HC catalysts (M=Ni, Cu, Ru, Ir). These discussions about generality of methods have added in revised manuscript (Page 19, highlighted in **RED**). The corresponding Supplementary Figs. 44-47 have been added in revised Supplementary Information.

Supplementary Figure 43 | Picture of ~1.22 g Fe-SAs/NPS-HC catalyst.

Supplementary Figure 44 | Characterizations of the Ni-SAs/NPS-HC catalyst. (a) TEM image of the Ni-SAs/NPS-HC. Scale bar, 500 nm. (b) XRD pattern of the Ni-SAs/NPS-HC catalyst. (c) HAADF-STEM image and corresponding EDS element maps (Ni: blue, C: red, N: green, P: orange, S: yellow). Scale bar, 100 nm. (d) AC HAADF-STEM image shows that isolated Ni single atoms are observed. Scale bar, 5 nm.

Supplementary Figure 45 | Characterizations of the Cu-SAs/NPS-HC catalyst. (a) TEM image of the Cu-SAs/NPS-HC. Scale bar, 500 nm. (b) XRD pattern of the Cu-SAs/NPS-HC catalyst. (c) HAADF-STEM image and corresponding EDS element maps (Cu: blue, C: red, N: green, P: orange, S: yellow). Scale bar, 100 nm. (d) AC HAADF-STEM image shows that isolated Cu single atoms are observed. Scale bar, 5 nm.

Supplementary Figure 46 | Characterizations of the Ru-SAs/NPS-HC catalyst. (a) TEM image of the Ru-SAs/NPS-HC. Scale bar, 500 nm. (b) XRD pattern of the Ru-SAs/NPS-HC catalyst. (c) HAADF-STEM image and corresponding EDS element maps (Ru: blue, C: red, N: green, P: orange, S: yellow). Scale bar, 100 nm. (d) AC HAADF-STEM image shows that isolated Ru single atoms are observed. Scale bar, 5 nm.

Supplementary Figure 47 | Characterizations of the Ir-SAs/NPS-HC catalyst. (a) TEM image of the Ir-SAs/NPS-HC. Scale bar, 500 nm. (b) XRD pattern of the Ir-SAs/NPS-HC catalyst. (c) HAADF-STEM image and corresponding EDS element maps (Ir: blue, C: red, N: green, P: orange, S: yellow). Scale bar, 100 nm. (d) AC HAADF-STEM image shows that isolated Ir single atoms are observed. Scale bar, 5 nm.

Figure R4 | Apparatus diagram for the preparation of Fe-SAs/NPS-HC catalyst.

- **Comment: Furthermore, there are several critical scientific questions to be addressed.**
 - 1. (1) Can the atomic ratio of Fe:N:S:P:C be controllable? (2) Which element is the key component? (3) How about comon the electrocatalytic performance?**
 - 2. Note that the pyrolysis proceeds at 900degree. In such a high temperature, why Fe, S, and P remains but Zn vaporize? The authors claimed that Fe is coordinated by N while there are no Fe-S and Fe-P contribution. Then, what are the chemical states of S and P? How can S and P withstand the high temperature pyrolysis even in the case of Zn vaporization?**

Response: We thank the reviewer and appreciate the suggestion. In order to more clearly demonstrate our response to the comments, we have carried out point-to-point responses, as follows:

Point-to-point Responses to the Comments:

Comment 1.1: Can the atomic ratio of Fe:N:S:P:C be controllable?

Response: The atomic ratio of Fe:N:S:P:C can be controlled by changing the amount of precursors during the synthesis process. We have prepared a series of catalysts with different content of Fe. The synthesis procedure of these catalysts is the same with that used for the synthesis of Fe-SAs/NPS-HC except with different amount of iron precursor. The detail synthesis procedure is described as follows:

Synthesis of Fe-SAs/NPS-HC, Fe-SAs/NPS-HC-1/8, Fe-SAs/NPS-HC-1/4, Fe-SAs/NPS-HC-1/2 and Fe-SAs/NPS-HC-2. Typically, for the synthesis of Fe-SAs/NPS-HC, bis(4-hydroxyphenyl) sulfone (325 mg) and phosphonitrilic chloride trimer (152 mg) were dissolved in 100 mL of methanol, marked as solution A. Then, the as-obtained powder of ZIF-8 (400 mg) was dispersed in methanol (40 mL), followed by injecting 1.73 mL methanol solution of iron (III) nitrate nonahydrate (10 mg/mL). Subsequently, the dispersion and N, N-diethylethanamine (1 mL) were respectively added into the solution A, following by stirring for 15 h. The precipitate was collected, washed and finally dried in vacuum at 80 °C for 12 h. Finally, the as-obtained powder was placed in quartz boat, and then maintained 900 °C for 3 h in a tube furnace with a heating rate of 5 °C min⁻¹ under flowing Ar atmosphere. The synthesis of Fe-SAs/NPS-HC-1/8, Fe-SAs/NPS-HC-1/4, Fe-SAs/NPS-HC-1/2 and Fe-SAs/NPS-HC-2 are the same as that of Fe-SAs/NPS-HC expect that the one-eighth of iron precursor, one-quarter of iron precursor, half of iron precursor and double iron precursor are used, respectively.

The TEM images of Fe-SAs/NPS-HC-1/8, Fe-SAs/NPS-HC-1/4, Fe-SAs/NPS-HC-1/2 and Fe-SAs/NPS-HC-2 reveal that these catalysts all display uniform hollow morphologies (Supplementary Fig. 28). And on obvious signals for metallic Fe species are detected (Supplementary Fig. 28). The content of Fe of these catalysts are determined by ICP-OES analysis and are listed in Supplementary Table 5, which indicates the atomic ratio of Fe of catalysts can be controlled. In addition, we can tune the atomic ratio of S and P of catalysts by adding extra precursors containing S element (bis(4-hydroxyphenyl) sulfone) or P element (phosphonitrilic chloride trimer) in pyrolysis process. This strategy works well, which have been proved by the successful synthesis of Fe-SAs/NPS-C (see details in **Methods** in the revised manuscript).

Supplementary Figure 28 | XRD patterns of the Fe-SAs/NPS-HC samples with different Fe loadings. Inset: The corresponding TEM images. Scale bar, 1 μm .

Supplementary Table 5 | The Fe content of Fe-based catalysts determined by ICP-OES analysis.

Catalysts	Fe-SAs/NPS-HC-1/8	Fe-SAs/NPS-HC-1/4	Fe-SAs/NPS-HC-1/2	Fe-SAs/NPS-HC-S	Fe-SAs/NPS-HC-2
Fe content (wt%)	0.25	0.47	0.81	1.54	2.73

- The synthesis process of Fe-SAs/NPS-HC with different Fe content has been described in the section of Methods in the revised manuscript (Page 23, highlight in **RED**). And the corresponding Supplementary Fig. 28 and Supplementary Table 5 have been added in the revised Supplementary Information.

Comment 1.2: Which element is the key component?

Response: Isolated Fe atom is the key component in Fe-SAs/NPS-HC catalyst for both alkaline and acidic ORR performances. In order to investigate the effect of single iron atoms on ORR performance, the iron-free N, P, S co-doped hollow carbon polyhedron (NPS-HC) catalyst and iron-based single-atom catalysts with different content of Fe (Fe-SAs/NPS-HC-1/8, Fe-SAs/NPS-HC-1/4, Fe-SAs/NPS-HC-1/2 and Fe-SAs/NPS-HC-2) were prepared. Firstly, we evaluated the correlation of the content of Fe with alkaline ORR performance. As shown in Supplementary Fig. 29, NPS-HC catalyst exhibits obviously lower alkaline ORR performance in term of much negatively shifted $E_{1/2}$ of 0.912 V, smaller kinetic current density (J_k , 1.26 mA cm⁻² at 0.85 V) and larger Tafel slope (64 mV dec⁻¹) compared to Fe-SAs/NPS-HC with $E_{1/2}$ of 0.912 V, J_k of 71.9 mA cm⁻² at 0.85 V and Tafel slope of 36 mV dec⁻¹, which clearly indicates the existence of single iron atoms plays the essential role in delivering excellent ORR performance. As shown in Supplementary Fig. 29a, as the content of Fe increases, the corresponding ORR polarization curves gradually positively shifted and the corresponding the values of $E_{1/2}$ gradually increase,

suggesting the enhanced ORR activity is attributed to the increase of the amount of Fe in catalysts. The rising kinetic current density and decreasing Tafel slope indicate that the crucial role of Fe component in accelerating ORR kinetics (Supplementary Fig. 29b and c). The similar results are verified when we assess ORR performance in acidic media. As shown in Supplementary Fig. 29d-f, iron-free NPS-HC catalyst shows poor ORR performance. With the increase of the content of Fe, the ORR activity and kinetics are gradually enhanced in term of better $E_{1/2}$, higher kinetic current density and smaller Tafel slope. These results suggest that the Fe component as isolated atoms play the key role in ORR performance in both alkaline and acidic media. To further reinforce the above conclusion, the control experiments are designed and performed. It is known that SCN^- ion has a strong affinity for Fe and can poison the Fe- N_4 coordination in catalyzing ORR. When 0.01 M KSCN is added in O_2 -saturated 0.5 M H_2SO_4 , the onset potential and half-wave potential exhibit obviously negative shifts, along with the visible decrease of diffuse-limited current (Supplementary Fig. 30a). Then we rinse this electrode with water and re-measured it in 0.1 M O_2 -saturated KOH. It can be seen that the ORR polarization curve gradually recovers to the pristine level as the LSV tests go on, which is ascribed to the recovery of the blocked Fe- N_4 sites owing to the dissociation of SCN^- ion on the Fe active centers in 0.1 M KOH (Supplementary Fig. 30b). These results unambiguously reveal that the Fe component is essential to display the outstanding ORR performance.

- The discussion about the effect of Fe content on ORR performance has been added in the revised manuscript (paragraph beginning in Page 12, highlighted in **RED**). The corresponding Supplementary Fig. 29 and Supplementary Fig. 30 have been added in the revised Supplementary Information.

Supplementary Figure 29 | Electrocatalytic ORR performance of Fe-SAs/NPS-HC samples with different Fe loadings. (a-c) ORR polarization curves (a), comparison of J_k at 0.85 V and $E_{1/2}$ (b), Tafel plots (c) of NPS-HC, Fe-SAs/NPS-HC-1/8, Fe-SAs/NPS-HC-1/4, Fe-SAs/NPS-HC-1/2, Fe-SAs/NPS-HC, Fe-SAs/NPS-HC-2 in alkaline media. (d-f) ORR polarization curves (d), comparison of J_k at 0.85 V and $E_{1/2}$ (e), Tafel plots (f) of NPS-HC, Fe-SAs/NPS-HC-1/8, Fe-SAs/NPS-HC-1/4, Fe-SAs/NPS-HC-1/2, Fe-SAs/NPS-HC, Fe-SAs/NPS-HC-2 in acidic media. The content of Fe is determined by ICP-OES as 0, 0.25 wt%, 0.47 wt%, 0.81 wt%, 1.54 wt% and 2.73 wt% for NPS-HC, Fe-SAs/NPS-HC-1/8, Fe-SAs/NPS-HC-1/4, Fe-SAs/NPS-HC-1/2, Fe-SAs/NPS-HC and Fe-SAs/NPS-HC-2, respectively.

Supplementary Figure 30 | (a) ORR polarization curves of Fe-SAs/NPS-HC before and after the addition of 0.01 M KSCN in 0.5 M H₂SO₄. (b) ORR polarization curves of SCN⁻ poisoned Fe-SAs/NPS-HC by continuous LSV tests in 0.1 M KOH.

Comment 1.3: How about the effect of the composition on the electrocatalytic performance?

Response: The effect of Fe composition on ORR performance has been discussed above. In addition, we discover that the electronic modulation of active center can be achieved by near-range coordination with nitrogen and long-range interaction with sulfur and phosphorus, which can effectively promote ORR activity and kinetics in both alkaline and acidic media. Firstly, we evaluated the effect of N, P, S elements on alkaline ORR performance. As shown in Fig. 5f and Supplementary Fig. 38, the doping of heteroatoms leads to different degree of enhancement of ORR activity in view of $E_{1/2}$: N-doped Fe-SAs/N-C (0.870 V) < N, P co-doped Fe-SAs/NP-C (0.881V) < N, P, S co-doped Fe-SAs/NPS-C (0.894 V). Regarding the ORR kinetics, Fe-SAs/NPS-C exhibit higher kinetic current density of 34.6 mA cm⁻² at 0.85 V than those of Fe-SAs/NP-C (19.7 mA cm⁻²) and Fe-SAs/N-C (11.7 mA cm⁻²). The better ORR kinetics of Fe-SAs/NPS-C is further confirmed by smaller Tafel slope of 51 mV dec⁻¹ compared with Fe-SAs/NP-C (54 mV dec⁻¹) and Fe-SAs/N-C (59 mV dec⁻¹) (Fig. 5g). **This strategy of electronic modulation of active center also applies to the enhancement of acidic ORR performance.** As shown in Supplementary Fig. 39, the N, P, S co-doped Fe-SAs/NPS-C exhibits obvious enhancement in activity with more positive $E_{1/2}$ of 0.764 V compared with N-doped Fe-SAs/N-C (0.666 V) and N, P co-doped Fe-SAs/NP-C (0.725 V). Specifically, the kinetic current density and Tafel slope of Fe-SAs/NPS-C is much better than those of Fe-SAs/N-C and Fe-SAs/NP-C, suggesting the doped P and S atoms can tune electronic structure of Fe-N₄ active center to boost the acidic ORR kinetics (Supplementary Fig. 40). Moreover, DFT calculations are carried out to elucidate the roles of electronic control of the metal centers by the doping of P, S atoms, the calculated free energy diagram on different doped types of Fe-SAs samples at 1.23 V (thermodynamic equilibrium potential of ORR) was presented in Supplementary Fig. 35. It can be seen that all reaction steps are down-hilled in the 4e⁻ reduction path on Fe-SAs/NPS-C, suggesting ORR process is more thermodynamically favorable on Fe-SAs/NPS-C. The last two electrochemical steps of Fe-SAs/N-C and Fe-SAs/NP-C are more endothermic than Fe-SAs/NPS-C at $U^{\text{RHE}} = 1.23$ V. The binding energy of Fe-SAs/NPS-C for the intermediates is

the weakest than that of Fe-SAs/N-C and Fe-SAs/NP-C, indicating that Fe-SAs/NPS-C exhibits the highest reactivity than Fe-SAs/N-C and Fe-SAs/NP-C in alkaline media. Furthermore, we investigate the catalytic behavior of these three catalysts under acidic conditions (Supplementary Fig. 36 and 37). At $U = 1.23$ V, the first two reduction steps on Fe-SAs/N-C are exothermic, and the last two steps are endothermic by 0.48 and 0.59 eV, respectively. When P and S are doped, only the second step ($\text{OOH}^* + 3\text{H}^+ + 3\text{e}^- \rightarrow \text{O}^* + 2\text{H}^+ + \text{H}_2\text{O} (l) + 2\text{e}^-$) is exothermic, while the other three steps are endothermic. On Fe-SAs/NPS-C, the last two steps are endothermic by 0.23 and 0.27 eV, respectively, which are smaller than Fe-SAs/N-C and Fe-SAs/NP-C, indicating that the ORR process of Fe-SAs/NPS-C is thermodynamically favorable.

In summary, the high content of atomically dispersed Fe-N₄ active center is essential for the excellent ORR performance. And the surrounding P and S atoms can modulate the electronic properties of metal center to enhance ORR activity and kinetics.

According to your comment, the new supplemental data and electrochemical tests have been added in the revised manuscript, as follows:

- To investigate the effect of Fe composition on ORR performance, the iron-based single-atom Fe-SAs/NPS-HC catalysts with different Fe content were prepared. Then, we assessed the correlation of the content of Fe with ORR performance in both alkaline media and acidic media. Furthermore, we carried out the control experiments to reveal Fe component is essential to deliver the outstanding ORR performance. These investigations and discussions have been added in the revised manuscript (paragraph beginning in Page 12, highlighted in **RED**). The corresponding Supplementary Fig. 29 and Supplementary Fig. 30 have been added in the revised Supplementary Information.
- To investigate the effect of N, P, S composition on ORR performance, we prepared N doped single-atom Fe sample (Fe-SAs/N-C), N, P co-doped single-atom Fe sample (Fe-SAs/NP-C) and N, P, S co-doped single-atom Fe sample (Fe-SAs/NPS-C) and evaluated their ORR performance. Moreover, we discover that the surrounding P, S atoms can modulate the electronic properties of Fe-N₄ active sites to enhance ORR activity and kinetics. And DFT calculations reveal that P and S atoms can donate electrons to single-atom metal Fe center to make the charge of Fe ($\text{Fe}^{\delta+}$) become less positive, which contributes to a weakened binding of the adsorbed OH species, resulting in the enhanced ORR kinetics and efficiency. In this new round of respond, we find this electronic modulation strategy can be appropriate for enhancement of acidic ORR performance. The corresponding electrochemical tests and DFT results have been added in the revised manuscript (paragraph beginning in Page 16, highlighted in **RED**). And the corresponding Supplementary Fig. 36, Supplementary Fig. 37, Supplementary Fig.39 and Supplementary Fig.40 have been added in the revised Supplementary Information.

Supplementary Figure 39 | ORR polarization curves of single-atom Fe catalysts with different doped types in O_2 -saturated $0.5\text{ M H}_2\text{SO}_4$.

Supplementary Figure 40 | Electronic control effect on acidic ORR performance by experimental measurements. (a) comparison of J_k at 0.85 V and $E_{1/2}$ and (b) the Tafel plots of single-atom Fe catalysts with different doped types.

Supplementary Figure 36 | Free energy diagram of ORR on different doped types of Fe-SAs samples at $U^{\text{RHE}} = 1.23\text{ V}$ under acid condition. The last two electrochemical steps of Fe-SAs/N-C and Fe-SAs/NP-C are relatively strong endothermic, suggesting that Fe-SAs/N-C and Fe-SAs/NP-C exhibit less reactivity than Fe-SAs/NPS-C.

Supplementary Figure 37 | Proposed reaction scheme with the intermediates having optimized geometry of Fe-SAs/NPS-HC toward acidic ORR. (C: grey, N: blue, Fe: orange, S: yellow, P: green, O: red, H: white)

Comment 2: Note that the pyrolysis proceeds at 900degree. In such a high temperature, why Fe, S, and P remains but Zn vaporize? The authors claimed that Fe is coordinated by N while there are no Fe-S and Fe-P contribution. Then, what are the chemical states of S and P? How can S and P withstand the high temperature pyrolysis even in the case of Zn vaporization?

Response: Thanks for the reviewer's comment. At one standard atmosphere, the boiling point of Zn metal is about 908 °C, while the boiling point of Fe metal is about 2750 °C. Because the pyrolysis temperature of 900 °C is closed to the boiling point of Zn metal, Zn metal will vaporize away along with the Ar flow at such high temperature (please refer to *J. Am. Chem. Soc.* **2008**, 130, 5390; *Adv. Mater.* **2015**, 27, 5010; *Angew. Chem. Int. Ed.* **2016**, 55, 10800.). It is worth mentioning that the clever construction of MOF@polymer accelerates the decomposition of ZIF-8, and facilitates the migration and diffusion of Zn species, which is verified by TGA analysis and XRD measurements. Therefore, the evaporation of Zn metal is accelerated and the vast majority of Zn vaporizes, as supported by XRD patterns of Fe-SAs/NPS-HC sample that no corresponding Zn species signals are observed. Because the boiling point of Fe metal is much higher than the pyrolysis temperature of 900 °C, Fe species can remain. The S and P can be introduced into carbon matrix to form stabilized C-S and C-P species during high-temperature thermal treatment under an Ar flowing atmosphere, which leads to S and P element can be remain during pyrolysis process. This phenomenon has been reported in many previously reported literatures (please refer to: *J. Am. Chem. Soc.* **2014**, 136, 14385; *Angew. Chem. Int. Ed.* **2016**, 55, 2230; *Angew. Chem. Int. Ed.* **2016**, 55, 13296.). Furthermore, we have carried out energy-dispersive spectroscopy (EDS) mappings and XPS measurements to confirm the existence of Fe, S and P elements in our Fe-SAs/NPS-HC catalyst. As shown in Fig. 1e, Fe, S and P elements are distributed uniformly over the entire architecture of Fe-SAs/NPS-HC catalyst. The corresponding Fe, P and S element signals have been detected by XPS analysis (Supplementary Fig. 15), which further confirms the presence of these elements. We have performed XPS analysis to reveal the chemical states of S and P atoms (Supplementary Fig. 4). The P 2p spectrum can be deconvoluted into two peaks located at 132.8 eV and 133.9 eV, indexing to P-C and P-O (please refer to *Adv. Mater.* 2015, 27, 5010; *Chem. Eur. J.* 2014, 20, 3106). The S 2p spectrum can be fit well with three peaks at 164.0,

165.2, and 168.3 eV, corresponding to 2p_{3/2}, 2p_{1/2} splitting of the S2p spin orbital (-C-S-C-) and oxidized S, respectively (please refer to *Carbon* 2015, 92, 327; *Nanoscale* 2013, 5, 3283.). To confirm the statement that Fe is coordinated by N while there are no Fe-S and Fe-P contribution, we have carefully examined the XAFS analysis and investigated many relevant literatures. From the Fourier transform (FT) curves, the distance of Fe-N shell is located at about 1.5 Å. The distance of the Fe-S coordination is at ~1.8 Å, which is higher than that of Fe-N coordination (~1.5 Å) (please refer to: *Angew. Chem. Int. Ed.* 2017, 56, 610-614.). Because the atomic radius of P atom is larger than that of S atom, the distance of F-P shell is longer than that of Fe-S shell. As shown in Fig. 3c, only one peak at the distance of ~1.5 Å in Fe K-edge EXAFS Fourier transform curve of Fe-SAs/NPS-HC is detected, ascribed to Fe-N coordination. And other higher shell peak at the distance of ~2 Å is not observed. The result confirms that there are no Fe-S and Fe-P contribution in the Fe-SAs/NPS-HC catalyst. Furthermore, we have carried out the analysis of Mössbauer spectra and hyperfine parameters of Fe-SAs/NPS-HC. The Mössbauer spectrum of Fe-SAs/NPS-HC was fitted with two magnetic sextet subspectra evolved at 78 K from the relaxational RT-spectra, which were assigned to Fe^{III}N₄ (S=1/2) and Fe^{II}N₄ (S=1) species, indicating that the Fe atoms are only coordinated by N atoms (please refer to: *Angew. Chem. Int. Ed.* 2017, 56, 13800–13804). However, no other Fe-related phases (e.g. Fe, Fe_xC, Fe_xS, Fe_xP) were observed. The findings are consistent with the XAFS analysis. All results are strong proofs to confirm that no Fe-S and Fe-P coordination structures in the Fe-SAs/NPS-HC catalyst.

Comment: Overall, I would suggest a further thorough revision to improve the quality of the manuscript.

Response: We appreciate the reviewer #2's comments. Herein, we have carefully addressed all the comments and suggestions and revised our manuscript. Many new data and contents have been supplemented to improve the quality and theme of our work and highlight the scientific novelty of our work. **The changes in the manuscript are marked in RED and the outline is listed as follows:**

- We have carefully revised our manuscript to demonstrate the advantages of our synthetic method compared with many conversional methods and highlight the novelty and significance of our work (paragraph beginning in Page 2, Line 8, highlighted in **RED**).
- To advance the scientific understanding on the ORR active sites at atomic scale, apart from *in situ* monitoring of electronic structure of active sites under ORR operating conditions by *in situ* XAFS measurement, we have supplemented DFT calculations to reveal the structural evolution of active sites with the intermediates during ORR process in both alkaline and acidic media. The corresponding Supplementary Fig. 35, Supplementary Fig. 36 and Supplementary Fig. 37 have been added in revised Supplementary Information. Furthermore, we carried out control electrochemical experiments to demonstrate Fe active center is essential to deliver the excellent ORR performance. The corresponding Supplementary Fig. 30 has been added in revised Supplementary Information. These discussions have been added in revised manuscript (Page 13 and Page 16, highlighted in **RED**).
- The large-scale preparation of catalysts is highly desirable for further commercial application. To demonstrate the mass production, grams of Fe-SAs/NPS-HC have been prepared, indicating our synthetic method is easily scaled up. This part has been added in revised

- manuscript (Page 19, highlighted in **RED**). The corresponding Supplementary Fig. 43 has been added in revised Supplementary Information.
- To demonstrate the generality of our synthetic method, we have extended our synthetic method to other metal single-atom catalysts, and we have achieved the synthesis of various M-SAs/NPS-HC catalysts (M=Ni, Cu, Ru, Ir). These discussions about generality of methods have added in revised manuscript (Page 19, highlighted in **RED**). The corresponding Supplementary Figs. 44-47 have been added in revised Supplementary Information.
 - The synthesis process of Fe-SAs/NPS-HC with different Fe content has been described in the section of Methods in the revised manuscript (Page 22, highlight in **RED**). And the corresponding Supplementary Fig. 28 and Supplementary Table 5 have been added in the revised Supplementary Information.
 - To investigate the effect of Fe composition on ORR performance, the iron-based single-atom Fe-SAs/NPS-HC catalysts with different Fe content were prepared. Then, we assessed the correlation of the content of Fe with ORR performance in both alkaline media and acidic media. Furthermore, we carried out the control experiments to reveal Fe component is essential to deliver the outstanding ORR performance. These investigations and discussions have been added in the revised manuscript (paragraph beginning in Page 12, highlighted in **RED**). The corresponding Supplementary Fig. 29 and Supplementary Fig. 30 have been added in the revised Supplementary Information.
 - To investigate the effect of N, P, S composition on ORR performance, we prepared N doped single-atom Fe sample (Fe-SAs/N-C), N, P co-doped single-atom Fe sample (Fe-SAs/NP-C) and N, P, S co-doped single-atom Fe sample (Fe-SAs/NPS-C) and evaluated their ORR performance. Moreover, we discover that the surrounding P, S atoms can modulate the electronic properties of Fe-N₄ active sites to enhance ORR activity and kinetics. And DFT calculations reveal that P and S atoms can donate electrons to single-atom metal Fe center to make the charge of Fe (Fe^{δ+}) become less positive, which contributes to a weakened binding of the adsorbed OH species, resulting in the enhanced ORR kinetics and efficiency. In this new round of respond, we find this electronic modulation strategy can be appropriate for enhancement of acidic ORR performance. The corresponding electrochemical tests and DFT results have been added in the revised manuscript (paragraph beginning in Page 16, highlighted in **RED**). And the corresponding Supplementary Fig. 36, Supplementary Fig. 37, Supplementary Fig.39 and Supplementary Fig.40 have been added in the revised Supplementary Information.

Generally speaking, in this work, we not only developed a novel and simple MOF@polymer strategy to simultaneously achieve the synthesis of unique atomically dispersed active sites, the construction of functionalized hollow structure via Kirkendall effect and electronic modulation of active centers by near-range coordination with nitrogen and long-range interaction with sulfur and phosphorus, but also fabricated a remarkable ORR electrocatalyst (Fe-SAs/NPS-HC) with excellent performance in both alkaline and acidic media. In addition, the new concept of material design via structure functionalities and electronic control of active sites at the atomic scale are developed and evidenced. We believe that this design concept can offer an important guideline for researchers to design and optimize catalysts for achieving highly efficient catalytic process. Moreover, we have significantly advanced the scientific understanding on the ORR active sites at

atomic scale through *in situ* monitoring of electronic structure of active sites under ORR operating conditions by *in situ* XAFS measurement. And we find that our catalyst exhibits a great potential application in the substitution of the expensive Pt-based electrocatalysts to drive PEMFC devices and Zn/air battery. And we think that our work will attract extensive research attention and promote the in-depth understanding in ORR active sites at atomic scale and the development of nonprecious metal catalysts in the application of energy conversion and storage devices.

Reviewer #3 (Remarks to the Author):

The authors answer all of my questions with satisfactory. I have no further questions and would like to recommend its publication in the current version.

Reviewers' Comments:

Reviewer #2:

Remarks to the Author:

In my opinion, the quality of the manuscript is greatly improved, through both supplemented new data and in depth discussion. Particularly, the importance of the work is now clearly indicated from mechanistic understanding to device application in PEMFC and zinc-air batteries. I believe the results presented in this revised manuscript will attract broad interest of readership. Therefore, I have no doubt in recommending the publication of the work at its present form.

Reviewer #3:

Remarks to the Author:

No further comments. I recommend its publication in Nat Comm in its current version.